# Physical, Mechanical and Transfer Properties at the Steel-Concrete Interface: A Review

Yousra Hachem [1], Mohamad Ezzedine El Dandachy [1] and Jamal M. Khatib [1,2,*]

[1] Faculty of Engineering, Beirut Arab University, Beirut 1105, Lebanon; yah222@student.bau.edu.lb or yousra.hachem@outlook.com (Y.H.); m.dandachy@bau.edu.lb (M.E.E.D.)
[2] Faculty of Science and Engineering, University of Wolverhampton, Wolverhampton DY3 3PX, UK
[*] Correspondence: j.khatib@bau.edu.lb

**Abstract:** The steel-concrete interface (SCI) is extensively acknowledged to affect the durability of reinforced concrete. The main objective of this paper is to conduct a state-of-the-art review that contributes to sufficient knowledge on the determination of the SCI properties and its effect on the overall performance of reinforced concrete elements. The physical characteristics at the SCI are influenced by segregation, flow, hydration, and drying shrinkage of concrete, hence affecting the presence of voids and cracks within this interface. The bond strength is one of the measures of the SCI and this is conducted through pull-out, push-in, and tie-beam testing. It was shown that the rebar shape and diameter, the anchorage length, the concrete grade strength, binder type (geopolymer concrete), and the distribution of aggregates have a significant effect on the interface properties and behavior, where geopolymer concrete offered improved bond behavior over conventional concrete. Various studies have demonstrated that the presence of the steel-concrete interface and the application of mechanical stresses contribute to the flow transfer (inflow/outflow) through the reinforced concrete structure. Some of these studies focused on the initial state of the SCI within the structure, and some conducted tests with shear loading on the SCI. Regarding the transfer properties at the SCI, it was shown that the presence of steel rebar, crack dimensions, degree of saturation of concrete, and the concrete mix design, influence the permeability of the concrete, specifically at the vicinity of the SCI, because of the development of micro-cracks at the interface. In other studies, the shear stresses were also found to affect the transfer properties through the SCI. Researchers have implemented several software solutions such as finite element models on ABAQUS and mesoscale numerical simulations and have used machine learning models that predict and verify the effects of bond failure behavior at the SCI. Good agreement was established between the numerical and actual experimental results. The influence of different exposure conditions on the steel-concrete interface that change throughout time needs to be dealt with, which includes moisture-related environmental conditions, variation in temperature, and chemical exposure. Furthermore, the influence of structural loading, such as "creep effect", deterioration (ageing) of material must be studied at the interface. The studies were limited to short-term behavior.

**Keywords:** steel-concrete interface (SCI); pull-out; push-in; tie beam; mechanical behavior; bond strength; permeability

## 1. Introduction

Reinforced concrete is one of the main composite elements that has been developed in the building sector. It has been commonly used in many engineering projects due to its lower cost of material and ease of construction [1,2]. The behavior of reinforced concrete structures is affected by the steel-concrete interface (SCI). Therefore, the SCI is a main factor to consider for steel-reinforced concrete structures [3]. The SCI has a substantial impact on steel corrosion and on the bond between steel and concrete. Steel corrosion occurs mainly due to the exposure of steel to environments such as chlorides. Furthermore, the quality and type of

steel and concrete used have an impact on the bond between them [4]. For example, Électricité de France (EDF) manages numerous reinforced concrete (RC) reactors in the country, and an assessment of the leakage rate of fluid at the SCI must be established in order to predict their long-term behavior and ensuring the safety of the structures. In concrete containment buildings (CCB) in nuclear power plants, there exist many interfaces which may form a preferential path for leakage within the structure [5–7]. The critical relationship between steel rebar and concrete is related to the structural behavior of these two materials, which is linked to the interfacial bond [8–10]. Thus, the bond mechanism between the steel and concrete in RC members must be considered in assessing the performance of RC structures [11,12].

The leakage rate of fluids within the reinforced concrete is significantly influenced by the properties of concrete, such as drying shrinkage and, most importantly, permeability [13–16]. A critical step toward comprehending the behavior of the leakage is defining and understanding all the physical characteristics in weak zones via cracks resulting from mechanical stresses and voids at the steel-concrete interface [17–20]. The physical characteristics of plain concrete that affect permeability have been extensively studied [11,13]. However, little research has been conducted about permeability on concrete reinforced with ribbed steel rebar. Hence, further research is required on the impact of the concrete physical characteristics and permeability on the steel-concrete interface of reinforced structures [21]. During the service life of reinforced structures, applied mechanical loadings will induce cracks in the concrete. Since the SCI is a weak zone, the cracks will detrimentally influence the flow of fluids in this zone, altering its passage and consequently affecting the permeability [22].

Previous researchers have studied local-physical characteristics that occur at the SCI between steel and concrete and their impact on the interface, but there is still a lack of results explaining the overall behavior that occurs at that interface [23]. This paper is a literature review that addresses the physical, mechanical, and transfer properties at the steel-concrete interface. This review will highlight the influence of mechanical shear loading at the interface, and mainly the influence of shear loading on the transfer properties will be discussed in terms of permeability with the penetration of gas or water within the interface. Furthermore, previously implemented models that predict and verify the effects of bond failure behavior at the SCI will be discussed.

## 2. Methodology

This paper is a literature review that displays an overview of the physical, mechanical, and transfer properties that occur at the steel-concrete interface. A systematic process was followed in order to identify relevant papers that are related to the topic. Using the Scopus database, keywords were searched to find related studies. The total number of collected research papers or records were 142, these records were screened to ensure that experimental studies related to the influence of mechanical loading on the steel-concrete interface and permeability/transfer of gas within this interface were included. After carefully reading the abstract of each of the 142 articles, unrelated records were removed. Hence, the total number of excluded articles related to corrosion effect was nine, which will not be addressed in this review. The remaining papers were assessed after reading the whole article, and another 17 papers were excluded. The total records remaining (116) were the total final studies included for the literature review (Figure 1).

The chosen papers varied among journal research articles, conference papers, and books. They were assembled according to mechanical behavior by studying the bond and gas percolation or fluid flow at that interface. The date of publication of each paper to be used was carefully taken into consideration and that is was related to recent performed studies. Full texts were screened, and data was then collected from the selected papers, summarized, and categorized according to the main sections.

This review consists of six main sections: Section 1: Introduction, in this section the topic is introduced, indicating the importance of conducting studies at the steel-concrete interface to acknowledge the physical properties, bond behavior, and permeability at the interface. Section 2: the methodology displays a description of how the articles were

collected and analyzed using several criteria for assessing the records and including them in this study. Moreover, in Section 3 an overview of the physical characteristics at the steel-concrete interface is presented, looking more into the effect of steel placing within the concrete on the porosity of concrete, crack formation when the structure is subjected to tensile loads, and the effect of binder inclusion as cement replacement; each of these factors are divided into Sections 3.1–3.3. These sub-sections are related as the presence of steel within the concrete affects the porosity between the steel bar and concrete, flexural cracks may be formed, and the inclusion of binder in the concrete may improve the concrete matrix with steel. Section 4 presents the effect of mechanical stresses on the interface bond using pull-out, push-in, and tie beam testing of reinforced elements. Section 4.1 displays the mechanical setup for the mentioned testing, and Section 4.2 discusses the main parameters, which are the geometry and diameter of the rebar, strength grade of concrete, anchorage length, and binder type. In Section 4.3 push-in testing is conducted to study the shear performance and transfer properties at the SCI, but Section 4.4 discusses the evolution of the applied tensile force related to the end displacement of the steel rebar in the longitudinal direction. Section 5 discusses the main factors (cracks, reinforcement, degree of saturation, and concrete mixture design) that influences the permeability at the SCI, while Section 6 emphasis the importance of the use of models in understanding the bond behavior at the steel-concrete interface. Finally, the last two sections include the research gaps and recommendations for future research indicating the main gaps in present works, and a conclusion that summarizes the main findings. The article's outline and important details are tabulated, as shown in Table 1.

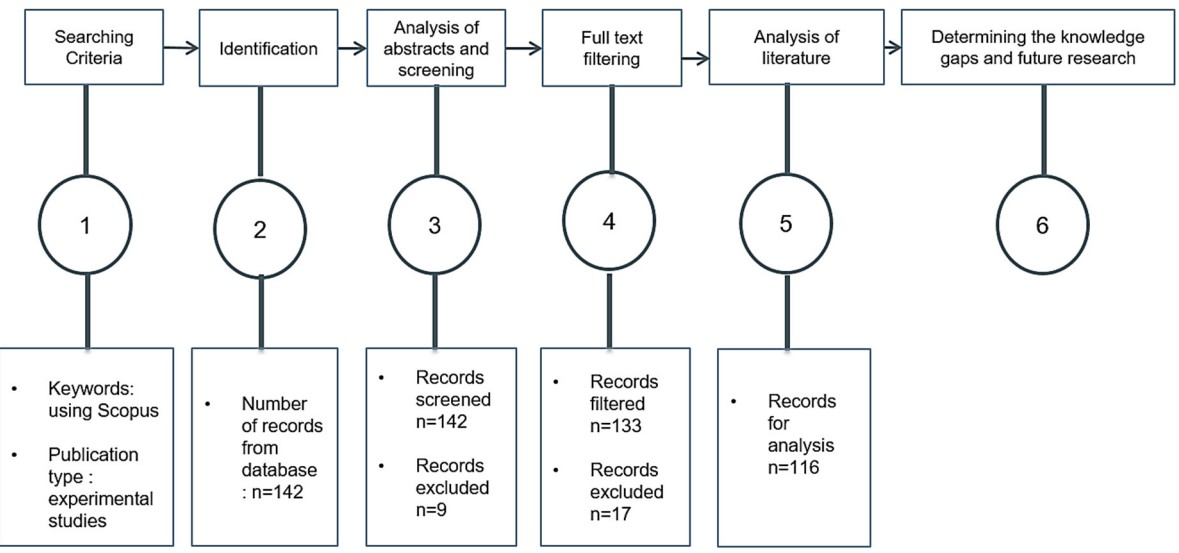

**Figure 1.** Flow chart of the review methodology.

**Table 1.** Summary of the manuscript outline.

| Section Number | Section Title | References | Recent | Context |
|:---:|:---|:---:|:---:|:---|
| 1 | Introduction | [1–23] | 2002, 2005, 2008, 2016, 2018, 2019, 2020, 2021, 2022 | Importance of studying the bond performance at the SCI |
| 2 | Methodology | - | - | Flow chart process for records and framing of the following sections |

**Table 1.** *Cont.*

| Section Number | Section Title | References | Recent | Context |
|---|---|---|---|---|
| 3 | Physical Characteristics at the Steel-Concrete Interface | [24–49] | 2004, 2005, 2006, 2007, 2011, 2017, 2020, 2021, 2022 | • Concrete porosity at the SCI<br>• Cracks at the SCI<br>• Binder effect at the SCI |
| 4 | The Effect of Mechanical Stresses on the SCI | [1–3,12,23,47–66] | 2017, 2019, 2021, 2022 | Bond strength at SCI testing methods pull-out, push-in, tie beam testing |
| 5 | The Permeability at the SCI | [23,26,40,57,60,67–106] | 1999, 2016, 2018, 2020, 2021, 2022 | Permeability at SCI testing methods pull-out, push-in, tie beam testing |
| 6 | Use of Models in Understanding the Bond Behavior at the SCI | [4,20,46,63,70,107–116] | 1984, 1999, 2001, 2010, 2011, 2012, 2018, 2020, 2021, 2022 | • Finite element modelling ABAQUS<br>• Mesoscale modelling<br>• Machine learning |
| 7 | Knowledge Gaps and Future Research | - | - | Present work's gaps and limitations |
| 8 | Conclusion | - | - | Summary of the main points based on the evidence presented |

## 3. Physical Characteristics at the Steel-Concrete Interface

### 3.1. Concrete Porosity at the SCI

The connected porosity in concrete is related to the aggregate interfacial transition zone [24,25], cement paste capillary pores [26,27], cement matrix, and aggregate microcracks [24,28]. The porosity of concrete can be influenced by several main factors, e.g., the particle size and location of aggregates, the water to cement ratio, crack configuration, voids, and the presence of reinforcing steel [4]. The relationship of concrete at the SCI can be compared to the relationship at the interfacial transition zone (ITZ), which is the region between concrete aggregate and cement paste. The ITZ's high porosity is due to constrained packing and rapid reacting cement grains with cement paste present in this zone and settling of the hydration products such as C-S-H and calcium hydroxide (CH) [23].

To compare the porosity at the SCI when steel is placed either vertically or horizontally (top or bottom), Horne et al. [23] detected high porosity for the smallest distance (μm) from the surfaces, which was accompanied by CH build-up of approximately 5 μm from both aggregate and steel surfaces, as shown in Figure 2. The steel-concrete interface porosity (between the steel bar and concrete), and the interfacial zone of bulk paste (between the cement paste and aggregate) were found to have similar behavior with the increase in the distance, indicating that the porosity at the SCI can be taken as the porosity at the ITZ [4]. The steel-concrete distance is defined as the distance between the exterior of the rebar and the closest concrete grain surface in contact with the concrete. However, the SCI thickness indicates the region of the same porosity between the concrete surrounding the steel rebar to achieve the same porosity as that of the bulk concrete. Figure 3 shows that the porosity at the SCI was maximum when it was closer to the rebar surface, and then porosity decreased when moving farther in distance [29]. This finding was consistent with Horne et al.'s results in [23]. Furthermore, Horne et al. [30] detected a coarser microstructure at the bottom of ribs of vertical rebar compared to other locations.

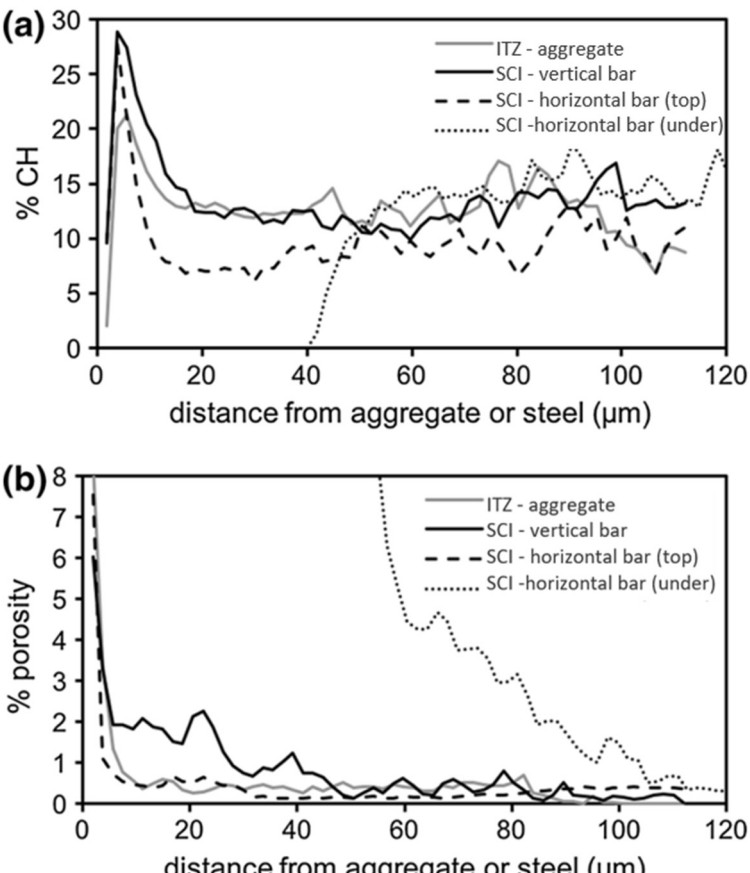

**Figure 2.** At the ITZ and SCI, (**a**) calcium hydroxide (CH) and (**b**) porosity percentage between steel rebar (9 mm) and cement. Duration 1-year w/c = 0.49 [23].

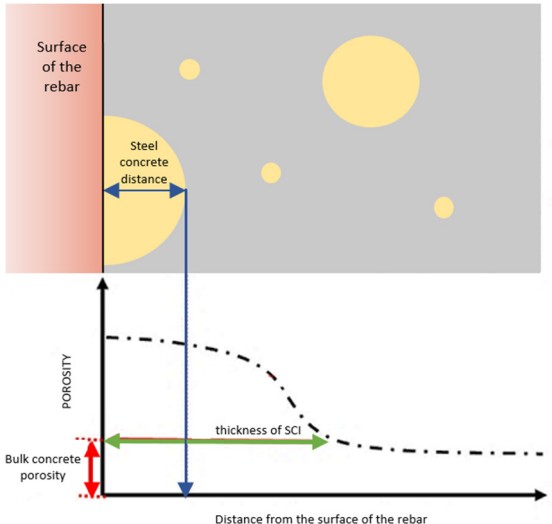

**Figure 3.** Effect of steel-concrete distance on the porosity and thickness of SCI (Reproduced with permission from reference [29]).

The concrete porosity controls the concrete quality in which it is related to the facility of fluid penetration in the concrete [31,32]. There exists a linear relationship between porosity and water absorption where when the capillary porosity increases the rate of water absorption (WA) relatively increases. The WA is influenced by the concrete position (e.g., top, middle, and bottom of the specimen). Khatib and Mangat [33] found that the top

surface of cubic specimens, which is of coarser porosity, absorbed 50% larger water amounts compared to concrete at the bottom surface, which is more compact. The difference in the water absorption at different locations can be explained where, during compaction, the sand and the coarse aggregates segregate and water moves upwards to the surface. This will cause variations in the pore size structure, porosity, and hydration of concrete.

At the SCI, air voids play a major physical role between the steel–fresh concrete and steel-void connection. The flowing direction of concrete during placement contributes to the extent to which it is compatible with the reinforcing rebar. Consequently, voids will form because less coarse aggregates and fresh concrete will reach the reinforcing steel bar [23]. In saturated concrete, the horizontal bar and bulk concrete influence the void size, which is around 100 mm [29]. The change in the microstructure will be due to the restrained drying shrinkage [21]. Angst [23] observed the voids using backscattered electron micrograph and the results have shown that bleed water zones under the rebar were approximately 200 μm, as shown in Figure 4. At the SCI, bleed water will gather under the steel rebar that can later be removed using shrinkage chemical admixtures. These voids are filled with water, so they are not entrapped air voids.

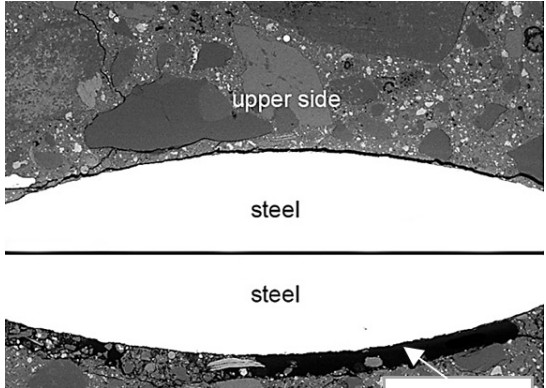

**Figure 4.** Zone of bleeding water under steel rebar, lower and upper SCI (PC: fly ash 20% w/b = 0.5) [23].

The increased presence of voids at the SCI due to poor compaction of fresh concrete is also related to the increase in the concrete thickness layer under the steel. The "top bar effect," known as a defect at the interfacial zone, demonstrates that voids are found under and along the reinforcement in the top part of structures [34,35]. As a result, the increase in porosity and macroscopic openings under bars placed horizontally is related to the top bar effect [36,37].

### 3.2. Cracks at the SCI

When reinforced structures are subjected to tensile loads (mechanical stresses), cracks and deformation in concrete occur, especially within the bond between the steel and concrete, thus affecting the microstructure. Upon reaching maximum loading, primary cracks that are wider than those present at the bar become visible. Furthermore, the transfer of load from steel to concrete initiates the formation of micro-cracks. Upon continuous loading, secondary cracks appear and detrimentally affect the SCI (Figure 5A). Another type of crack noted as longitudinal cracks propagate until reaching the exterior of concrete as shown in Figure 5B [38–40]. The different types of cracks accompanied by outward loading led to strength loss at the SCI, such as separation of concrete and bar slip. When the concrete surface or cover separates from the rebar, this is referred to as "separation", whereas "slip" occurs when the bar slides horizontally through the concrete. The crack width is determined by the rebar diameter, reinforcement ratio, bonding properties, concrete strength, and cover thickness [41].

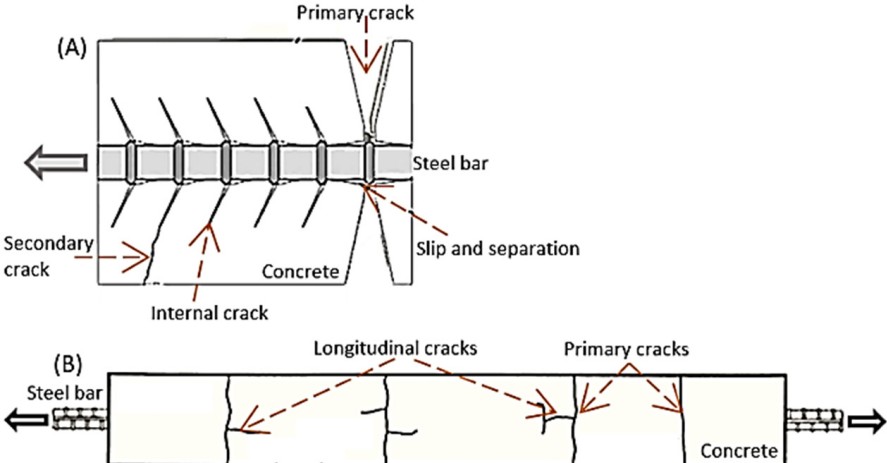

**Figure 5.** Detailed representation of (**A**) concrete cracking (primary, secondary, internal concrete cracking) and (**B**) primary then longitudinal cracking due to tensile stresses (Reproduced with permission from reference [38]).

### 3.3. Binder Effect at the SCI

The binder type may be a crucial factor in affecting the concrete as it is a component of the SCI due to variation in the cement grains' packing, reactivity, and hydration products created. Alkali-activated geopolymer mixture has received a lot of interest since it does not incorporate Portland cement binder, where hydrates may be formed by depolymerization or polycondensation reaction involving alkaline activators such as sodium silicate ($Na_2SiO_3$) and sodium hydroxide (NaOH) solutions with aluminosilicate minerals such as metakaolin, fly ash, slag, red mud, and steel slag [42–44]. Previous research has shown that fly ash-based geopolymer concrete (FGPC) offers various advantages over conventional concrete, including greater tensile and compressive strengths [44]. As a result, geopolymer may be used as a sustainable alternative material to cement, thereby lowering cement usage in the construction sector, and it is environmentally friendly [45,46]. The main difference between normal concrete and ferronickel slag and blast furnace slag powder (F-S powder) is the composition and amount of their binder components, which might result in a distinct bleeding capacity in concrete. Liu et al. [47] prepared a cement paste in order to study the impact of partial cement substitution by F-S powder (0%, 10%, 20%, 30%, 40%, and 50%) on bleeding capacity. The characteristics of the concrete were examined using the scanning electron microscopy (SEM) technique, and results showed that replacing the cement with FS powder increased the bleeding capacity in a binder material, which will, respectively, allow the formation of a large void underneath the rebar, which will also diminish the bond quality between steel and concrete [47]. To offset the effect of bleeding, it is recommended to lower the water to binder ratio, which will also help in reducing the void size.

Rasoul et al. [48], in their investigation, revealed that the presence of heavy metals in copper slag delayed the cement hydration, where copper slag might delay the setting time of the concrete mixture, however the concrete mechanical properties still surpassed the criteria when the copper slag replacement ratio was less than 50%. Sharma et al. [49] found that the low water absorption and smooth surface of copper slag might result in the presence of a large significant amount of excess water in the mixture, potentially causing the deterioration of the concrete's mechanical properties, but the quality of concrete could be managed by altering the concrete mixture proportion and the copper slag replacing ratio. Hence, the copper slag material replacement ratio can be raised and the superplasticizer content can be reduced as the smooth surface of copper slag material increases the fluidity of the concrete [50]. After exposure to high temperatures, mixing copper slag will induce more micro-cracks in concrete, deteriorating the concrete resistance to penetration by chloride ions and causing corrosion of steel bars embedded in concrete [49].

Experimental results of geopolymer-based ultra-high-performance concrete (G-NSC) specimens, constituted of flake sodium hydroxide (NaOH) and sodium silicate ($Na_2SiO_3$) solutions used as the alkaline activator, showed a greater bonding behavior with the steel bar. This was most likely owing to the geopolymer's non-polar polymer network structure, which consisted of aluminum-oxide and silicon-oxygen tetrahedrons. This network structure was compared to "microfibers", which might improve the concrete cracking and tensile resistance and absorb greater energy. Furthermore, the alkaline activator included into geopolymer concrete had a substantial impact on the cementitious materials. The alkaline activator may activate and increase the hydration of cementitious materials, hence improving bonding between concrete matrix and aggregate, efficiently limiting the expansion of micro-cracks in the concrete and leading to improved bonding performance with steel rebar [44].

In summary, the physical properties mainly related to the properties of the steel used, whether ribbed or not, and the concrete mixture components when binders are added strengthens the concrete matrix in comparison to plain concrete, which also leads to higher average bond strength indicated by results from studies conducted by Liu et al. and Rasoul et al. These factors will impact the segregation, flow of concrete, hydration, and drying shrinkage of concrete, hence leading to the presence of voids and cracks within this interface. Hence, porosity and cracks are formed at the contact surface of the reinforcing steel with the concrete.

## 4. The Effect of Mechanical Stresses on the SCI

The steel-concrete interface bond strength is a function of several bond mechanisms. The main three forces are friction resistance force along with the whole interface of the steel and concrete, force due to the mechanical interconnection of an uneven interface known as mechanical interlocking, and chemical bond force or adhesion between the steel outer coating and the concrete substrate [3,56]. Furthermore, there are two stages that describe the failure mode of the pull-out reinforced specimens: the working stage (crack) and the elastic stage (no crack) [61].

### 4.1. Mechanical Loading Test Setup

Improvements in the steel pull-out resistance can be achieved by managing the development of cracks due to the material properties. In the pull-out system shown in Figure 6, the specimens under study include ribbed reinforcing steel of different diameters of 16 mm and 20 mm and smooth steel of 8 mm with a variation in the C30 and C50 strength grades of concrete, in addition to two different anchorage lengths of 200 and 300 mm. The specimens were loaded at different increments, first at 2 to 5 kN and then at 5 to 10 kN. The displacement was then recorded [1].

The push-in test was conducted on steel reinforced cylindrical specimen G1 to evaluate the displacement of the top steel rebar, i.e., ribbed under applied load as shown in Figure 7 [57]. The type of failure that must occur within the specimen is required to be shear failure at the SCI, as per the chosen specimen configuration. Percolating gas, such as nitrogen, under pressure gradients across the specimen in the longitudinal direction is used to assess gas conductivity [57]. The specimen had a compressive strength of 28 MPa and was reinforced with a 16 mm diameter steel rebar. The anchorage length was within the interface and was set to be 7 cm according to Tixier to reach a shear failure at the SCI [58].

Tie beam testing was conducted on normal strength concrete (NSC) and fiber reinforced concrete (FRC) [59,60]. A steel rebar of 11 mm diameter and supporting concrete in an RC structure subjected to tensile stresses comprise the tie specimen, as shown in Figure 8. The cross section of the sample is 90 mm × 90 mm with a thickness cover of 40 mm. The displacement rate is set to 0.05 mm/min. The structure assures that only the tie-specimen experiences yielding of the rebar. The placement of gauges is to measure the strain.

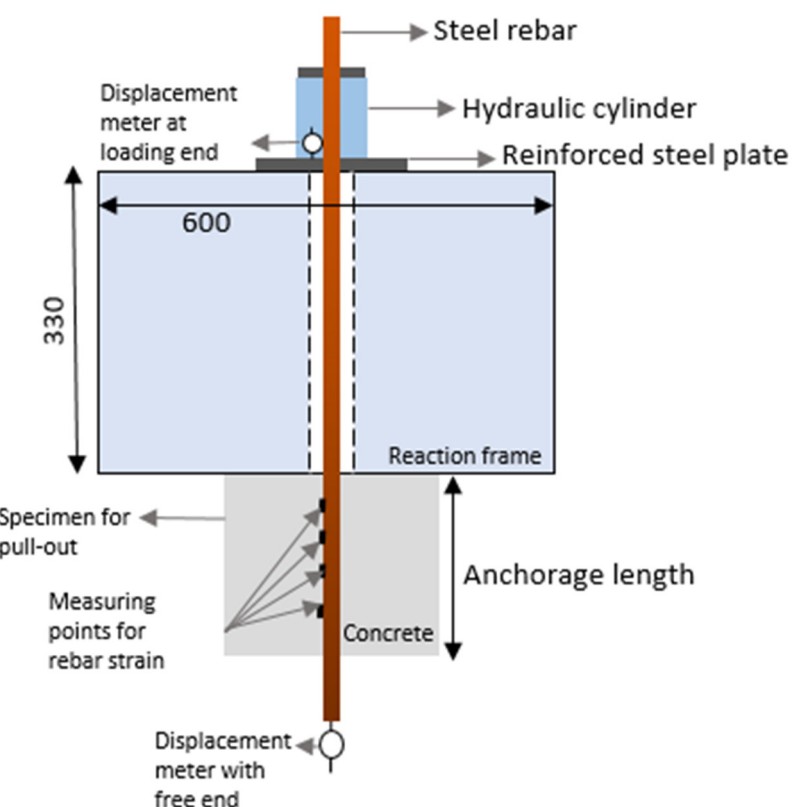

**Figure 6.** Pull-out loading device to study the pull-out resistance between steel and concrete at the interface (SCI) [1].

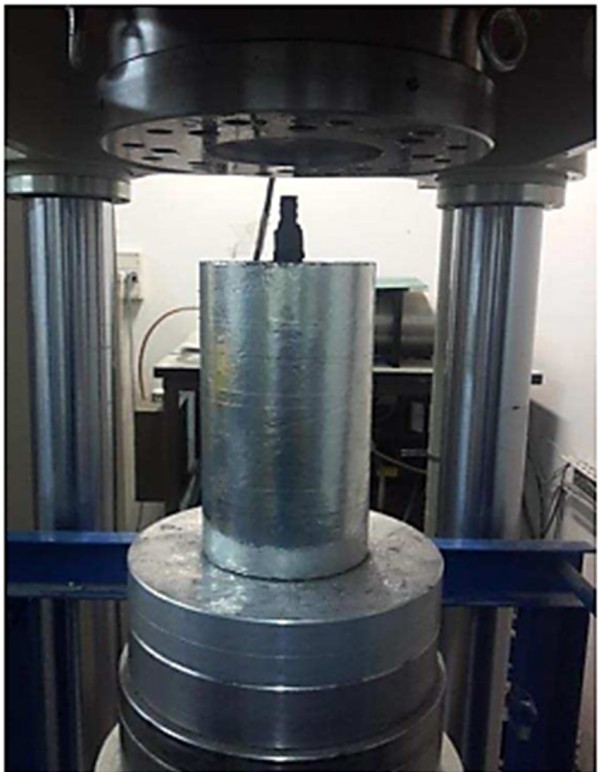

**Figure 7.** Push-in loading setup (adapted with permission from reference [57]).

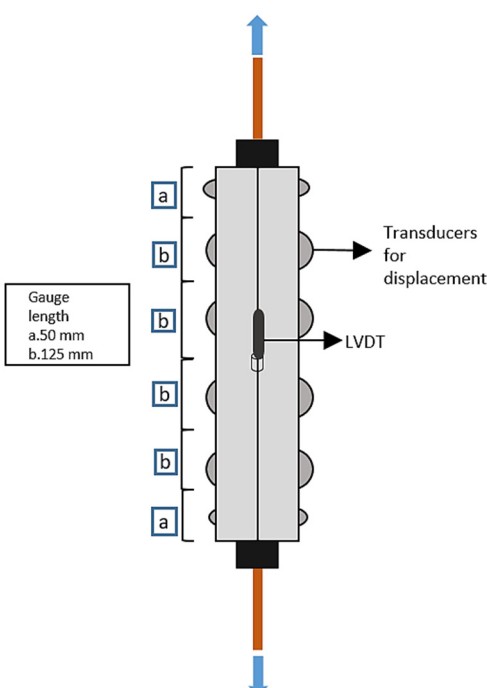

**Figure 8.** Tensile testing instrumentation for tie beam specimen (Reproduced with permission from reference [59,60]).

### 4.2. Mechanism Analysis of Interface Bond in a Pull-Out Setup

Researchers have conducted pull-out tests to verify the main physical factors that affect both the ultimate load of the bond interface and the bond strength of a reinforced concrete structure. Some of these factors are interrelated in which more than one factor will impact the bond at the SCI. The various factors affecting the bond at the SCI are described below.

#### 4.2.1. Geometry and Diameter of Reinforcing Steel Bar

Rebar shapes have evolved over time. Initially, the bars had a smooth squared or rounded cross-section [23]. It was later formed into asymmetrical shapes, such as ribbed steel bars, to strengthen the bond at SCI. In ribbed steel concrete structures, when chemical adhesion fails, mechanical interlocking takes place for bond strength at the interface [3]. Therefore, SCI is affected by the steel ribs' positions [23]. To ascertain whether interlocking occurs for smooth rebar in concrete, Ryota et al. [12] conducted experiments, and the results showed that for smooth rounded rebar the bond strength at the interface was low when compared to that of ribbed steel and concrete. This difference can be explained by the fact that interlocking does not exist for rounded rebar, so interlocking was not the main factor which contributed to the bond strength. Furthermore, Lei et al. [3] found that plain rebar did not significantly impact the bond–slip behavior because chemical adhesion simultaneously disappears at the interface when the interface undergoes slight slip. The effectiveness of the passive layer, which is formed around the steel bar due to corrosion, is degraded as this passive layer is seriously damaged under compressive stresses, unlike under tensile stresses [23]. Furthermore, the properties of the bond at the interface difference can be explained by the fact that interlocking does not exist for rounded rebar, so interlocking was not the main factor which contributed to the bond strength in this case.

To better understand the bond at the SCI, the results showed that in the pull-out test of the cylindrical specimen, the maximum bond stress was nearly independent of the bar diameter based on experimental work and simulations using different diameters [2]. Zhijian et al. [1] observed that specimens with ribbed steel rebar of 20 mm indicated an 8.5 times higher ultimate load value than that of smooth steel, as shown in Table 2 for specimens of the same C50 concrete strength and anchorage length of 300 mm. Furthermore,

it can be predicted that around 88% of the bond strength at the steel-concrete interface is accounted for by mechanical interlocking force [1].

**Table 2.** Ultimate loads in kN for pull-out reinforced concrete specimens of concrete strength grades of 30 and 50 (C30–C50), rebar diameter of 12, 16, and 20 mm, and anchorage lengths of 200, 300, and 400 mm [1].

| Specimen Number | Ultimate Load (kN) |
|---|---|
| C30-12-200 | 59.5 |
| C30-12-300 | 64.75 |
| C30-16-200 | 60 |
| C30-16-300 | 106.82 |
| C50-12-200 | 72.5 |
| C50-12-300 | 68.9 |
| C50-16-200 | 125.1 |
| C50-16-300 | 128 |
| C50-16-400 | 119.7 |
| C50-20-300 | 153.77 |

The friction in bond stress cannot be evaluated for concrete and ribbed steel, so the impact of friction coefficient $\mu$ was studied between the concrete and steel rebar. Mengjia et al. [63] found that the bond strength increased linearly with the increase in friction coefficient, but after a certain value for friction coefficient of 0.3, the bond strength was constant (Figure 9). Hence, no additional damage has occurred at the interface. This result can be explained by the fact that the concrete crushing occurred before the compressive strength of concrete was achieved [2].

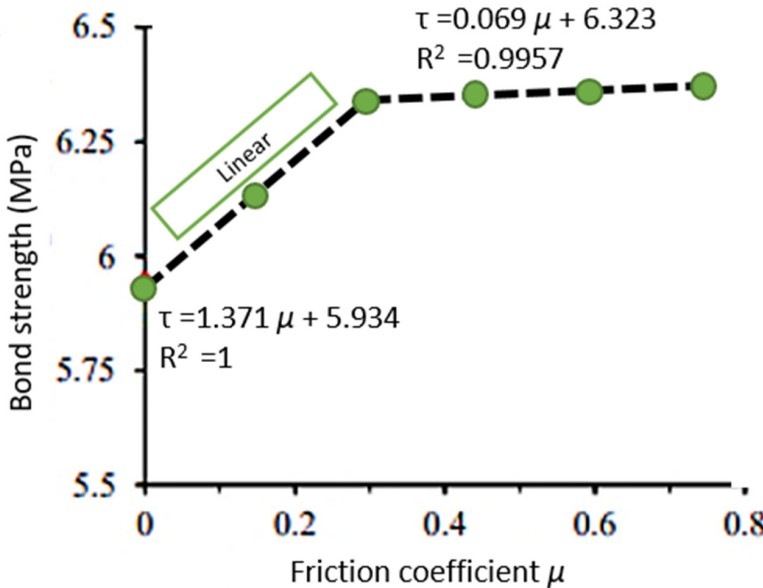

**Figure 9.** The effect of friction coefficient $\mu$ on the bond strength of concrete [63].

4.2.2. Strength Grade of Concrete

The strength grade of concrete is influenced by the mix proportions of cement, water, sand, and aggregates. An increase in bond strength in Figure 10 was observed for the different specimens; this increase was dependent on the higher concrete compressive strengths. The higher the strength grade of the concrete, the stronger the bond will be between the concrete and reinforcement, since the homogenous structure of concrete provides a better surface area for the reinforcement to adhere to [46,53]. Therefore, the

strength grade of concrete must be carefully considered in the design of reinforced concrete structures to ensure that the bond strength between the concrete and reinforcement is sufficient to withstand the anticipated loads and stresses. The anchorage length had no effect in this case. Moreover, the mode of failure changed from pull-out of steel bar to tensile failure with the increase in concrete strength [1].

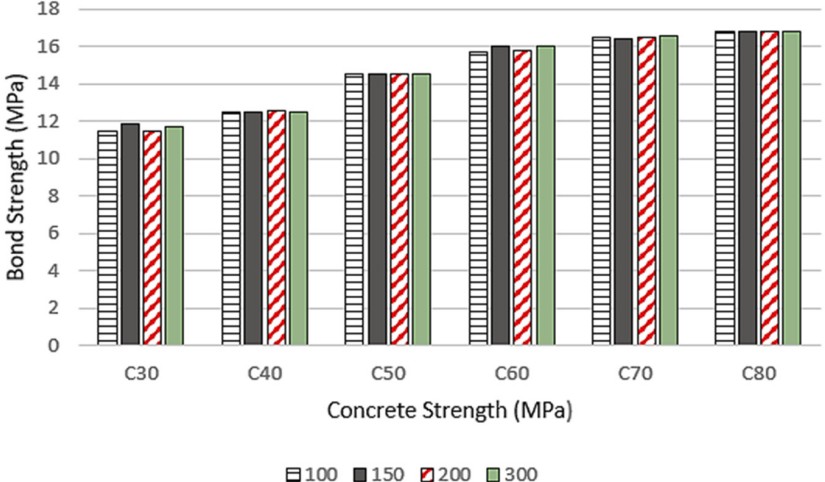

**Figure 10.** The effect of concrete strength on bond strength investigated with RC specimens (rebar diameter d16 of 16 mm and anchorage length ranging from 100 to 300 mm) [1].

Lei et al. [3] showed that the bond strength at the interface for deformed bars increased as the concrete strength increased. In the bond–slip curve, the ascending stage of the slope indicates the bond stiffness, which is related to the interface's resistance to slip. In Figure 11, it is evident that confined specimens have a significantly enhanced interface bond stiffness when compared to unconfined specimens. However, a softening stage is visible in the bond–slip curve for the confined specimens with deformed bar since the failure mode changes from splitting to splitting and pull-out post confinement. Moreover, in the specimen, the bond stress during the softening stage can sustain a higher level and is positively correlated with concrete strength. Under the pull-out test, the increase in bond strength was related to the interface bond strength. In this case, mechanical interlocking affected the interface bond where the concrete and ribs of the bar acted together. As a result, the strength of concrete influences the bond characteristics at the steel and concrete interface.

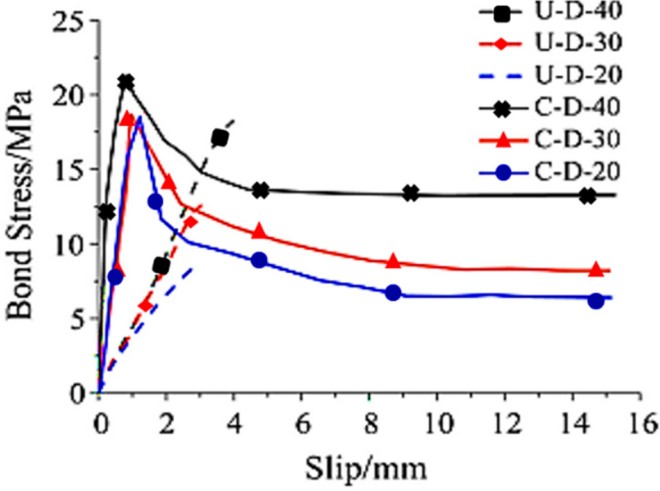

**Figure 11.** Bond–slip behavior of the different steel rebars. (U or C) for unconfined or confined, (D) for deformed bar, (20, 30, 40) for concrete strength grade (Adapted with permission from reference [3]).

### 4.2.3. Anchorage Length

The anchorage length is defined as the reinforcing steel rebar length embedded in concrete. The anchorage length is almost equal to the length of the rebar and has different values depending on the study. There is a direct relationship between the anchorage length and the pull-out resistance at the SCI. In Figure 12a, the bond strength did not vary with the increase in anchorage length from 100 to 300 mm; therefore, no effect on the bond strength has been recorded. Figure 12b shows that for anchorage lengths varying from 100 to 200 mm, the ultimate load increased until it reached a maximum of 75 kN. The increase in ultimate load was recorded by 64.2% for 200 mm of anchorage length compared to specimens with 100 mm of anchorage length. Beyond that, the ultimate load remained constant with the increase in anchorage length. For shorter anchorage lengths, the ultimate load increased with the increase in anchorage length, and the failure mode shifted from pull-out failure into tensile failure of the specimen [1].

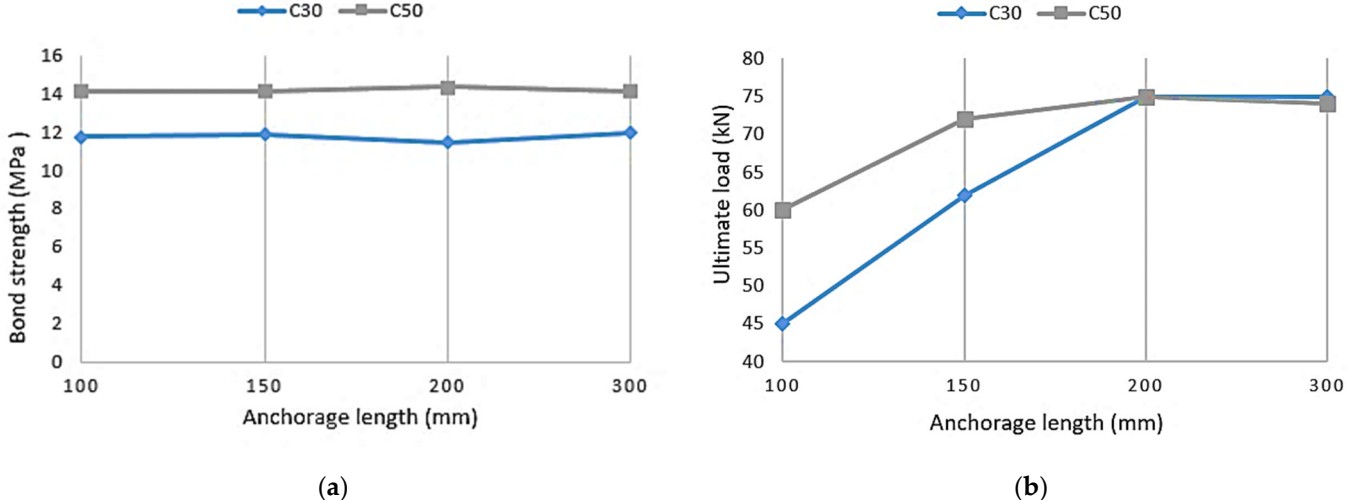

(**a**)                                               (**b**)

**Figure 12.** The effect of anchorage length on the (**a**) bond strength (**b**) ultimate load [1].

Yuxin et al. [81] observed that with the increase in anchorage length, the presence of voids increases relatively. Therefore, it is necessary to add a material such as rubber pads to decrease the ability of fluid or gas flow between the steel and concrete. Nevertheless, Chu and Kwan [62] found that stiffness and bond strength decreased when inserting a soft rubber pad at the interface.

The anchorage length is a main physical factor that affects the ultimate load and bond strength at the SCI. When designing the anchorage length, design codes play a significant role in specifying the required anchorage length. In design codes, i.e., ACI-318-11 and AS3600, the anchorage length will be large, leading to high construction costs and difficulty in implementation for structures that are prefabricated. Hence, the anchorage length must be taken as 0.5 times the design value of the JTG 3362-2018 code with the consideration of 1.7 times as a factor of safety [64–66].

### 4.2.4. Binder Type

Significant research on geopolymers has been carried out in order to address the formulation of geopolymers utilizing materials (binder) from various sources in order to obtain appropriate mechanical properties [51,52]. The potential bond performance of concrete mixtures with partial replacement of Portland cement with other cementitious materials (fly ash, slag, metakaolin, etc.) was essentially conducted through direct pull-out testing and researchers established substantial outcomes. Xuhong et al. [47] performed pull-out tests with specimens of concrete strength grades C30 and C35 with ferronickel slag and blast furnace slag powder (F-S powder) replacement percentages of 0%, 10%,

20%, 30%, 40%, and 50%, and reinforced with a ribbed rebar. The bonded and debonded lengths of the rebar were both 5d, where d is the ribbed rebar nominal diameter. The results showed that for smaller F-S powder content the bond strength was comparable to that of normal concrete. However, the increase in the F-S powder content has adversely affected the reinforcement bond quality. In this experiment it can be explained that the more F-S content, the lower the bond strength compared to that of normal reinforced concrete is [47]. The bond strength of a cubic concrete specimen (150 × 150 × 150 mm) with a ribbed steel bar of 14 mm in diameter was investigated by Chen et al. [50] The different mixtures were prepared using the same properties, but with various copper slag replacement ratios of 0, 20, and 40%. The bond length between the concrete and the steel rebar was adjusted to 70 mm, that is five times the steel rebar diameter. It was found that the bond strength of normal concrete at ambient temperature was higher than that of mixtures with varied copper slag replacement ratios of 20 and 40%. It was also found that the bond strength decreased with the increase in the copper slag replacement ratio, which might be attributed to bleeding-induced micro-cracks between the copper slag aggregate and cement paste. Nevertheless, as the heating temperature rises, the bond strength significantly decreases; this might be attributed to an increase in concrete compressive strength generated by the cement paste rehydration reaction under high temperature related to the steam curing effect [50]. Castel and Foster [53] studied the bond strength of geopolymer concrete (fly ash based) with slag as a partial substitute for fly ash. It was found that the bond strength in comparison to ordinary Portland concrete was slightly lower and slightly higher for plain rebar and ribbed rebar, respectively. Furthermore, Fernandez et al. [54] investigated the geopolymer concrete (fly ash based) bond performance in comparison to ordinary Portland cement. The variables of the different specimens were the diameter of the rebar and the type of the alkaline solution. In general, ordinary Portland concrete established a lower bond strength than the geopolymer concrete. For larger bar diameter a breakage in the geopolymer matrix was observed, and for smaller bar diameter the rebar rupture occurred before slipping for geopolymer concrete [53,54]. Bond strength of fly ash-based concrete to steel rebar is an important criterion for assuring the durability and safety of reinforced concrete components. Geopolymer concrete of high strength resulted in improved friction forces and chemical adhesion at the SCI. The increase in the fraction and length of steel fibers improved the shear and tensile properties of geopolymer concrete. The bond interface became rougher, which increased mechanical interlocking and friction resistance, enhancing the bond stiffness and performance while restricting the slip between the geopolymer concrete and ribbed steel rebar [44].

Albidah et al. [55] studied the effect of steel fibers on the bond strength of geopolymer concrete with and without steel fibers, and different sodium silicate to sodium hydroxide ratios of 0.6 and 1. Two modes of failure were established for geopolymer concrete specimens reinforced with steel fibers (0.5 mm in diameter and 30 mm long) and embedded with ribbed steel rebar of 12 mm diameter: splitting of the concrete and rebar pull-out. The fiber reinforced specimen with $Na_2SiO_3$ to NaOH ratio of 0.6 failed by pull-out of the rebar, whereas the steel fiber reinforced specimen with ratio of 1 failed with the concrete splitting with a bond strength of 6.2 MPa. Figure 13 presents the average bond strength for the plain and 200 mm fiber-reinforced geopolymer concrete cube with embedded steel rebar, and bonded length of rebar as 5d (five times the diameter of the rebar). It can be observed that the addition of steel fibers into the geopolymer concrete mix enhanced the bond strength at the SCI. The percentage increase in average bond strength varied from 7% to 45%. In particular, the inclusion of steel fibers changed the failure mode from concrete splitting to bar pull-out. The difference in the failure modes is due rather to the steel fiber's high splitting tensile strength in the geopolymer concrete matrix in comparison to plain concrete, which also leads to higher average bond strength. The mechanical resistance of reinforced concrete with concrete mixes of different binder types and under pull-out testing is summarized in Table 3.

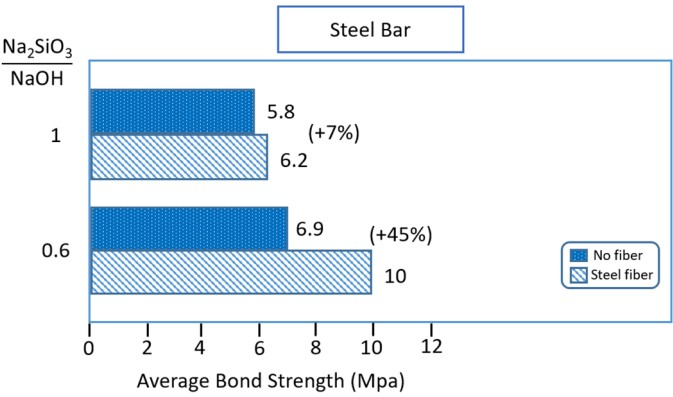

**Figure 13.** Effects of Na$_2$SiO$_3$ to NaOH ratio and steel fiber inclusion on the bond strength of geopolymer concrete with embedded steel bars (Reproduced with permission from reference [55]).

**Table 3.** Summary of the mechanical resistance of reinforced concrete related to the binder type.

| Binder Type | Mechanical Resistance |
|---|---|
| Ferronickel slag and blast furnace slag powder (F-S powder) | • More F-S content → the lower is the bond strength compared to that of normal reinforced concrete |
| copper slag (replacement ratios of 20 and 40%) | • The bond strength of normal concrete at ambient temperature was higher than that of mixtures with varied copper slag replacement ratios<br>• The bond strength decreased with the increase in the copper slag replacing ratio |
| Slag as a partial substitute for fly ash | In comparison to ordinary Portland concrete<br>• Slight low bond strength using plain rebar and slag<br>• Slight high bond strength for ribbed rebar and slag |
| Fly ash | Geopolymer concrete<br>• High bond strength of geopolymer concrete compared to ordinary Portland concrete (OPC)<br>• Larger bar diameter → breakage in the geopolymer matrix<br>• Smaller bar diameter → rupture of rebar before slipping |
| Steel fibers | • Rough bond interface<br>  → increased mechanical interlocking<br>  → increased friction resistance<br>  → enhancement of bond stiffness<br>  → Restricting the slip between the geopolymer concrete and ribbed steel rebar. |

### 4.3. Mechanical Stresses of Push-In Test for Ribbed Steel

The push-in test [57] was conducted for ribbed steel-reinforced concrete specimen G1 with an anchorage length of 7 cm to study the shear performance and transfer properties at the SCI. This anchorage length is considered to be short compared to the total length of the specimen. The steel at the ends is free of displacement and the only reason that limits the sliding of the rebar is the 7 cm bond with the concrete. The specimen configuration was set so that only shear failure will occur at the SCI. For larger anchorage lengths, the failure would occur due to splitting of concrete along the length of the specimen. According to the push-in test, at the pre-peak phase, for smaller displacement values there is a contribution of chemical adhesion. Then, upon the increase in the applied loading (at the top part of the ribbed steel rebar) the displacement increased, and mechanical interlocking took place.

Hence, shear stresses are transmitted through the interface and local sliding of the rebar occurs. In this phase, a linear behavior is observed. At the peak load, the interface cannot withstand the transmitted shear stresses and failure occurs. In Figure 14, the maximum force and displacement recorded for specimen G1 were approximately 62 kN and 2.7 mm, respectively [57]. In the post-peak phase, the steel-concrete interface was not capable of transmitting additional stresses, which indicates the decrease in load. In this case, friction forces dominate with the global sliding of the rebar. After that, for larger displacement values beyond 52 mm, the friction force diminishes till reaching nearly 0 kN (no interface).

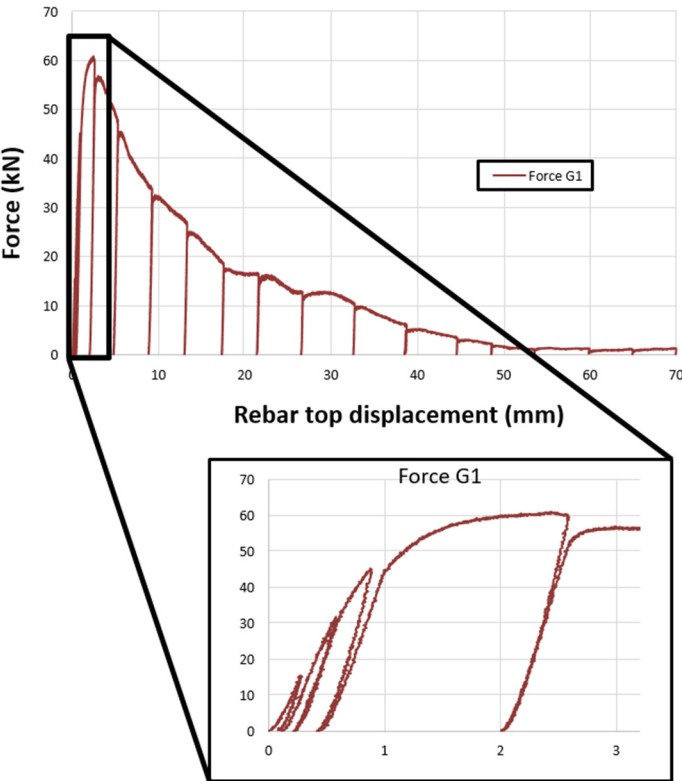

**Figure 14.** The SCI mechanical behavior under push-in testing (Adapted with permission from reference [57]).

*4.4. Mechanical Behavior and Displacement for Tie Beam Test*

Tensile testing in a tie-beam setup was conducted on normal strength reinforced concrete (NSC) and the results of the evolution of the applied tensile force versus the end displacement of the steel rebar in the longitudinal direction are presented in Figure 15 [60]. In the elastic (linear) phase and for loads starting from 0 to 15 kN, micro-cracks did not appear. Upon the increase in loading, between 15 and 25 kN, a non-linear behavior can be observed, and micro-cracks are initiated and propagated in the concrete. Moreover, there was a local development of macro-cracks with a further increase in loading in the loading phase for forces varying from 25 to 45 kN. Furthermore, it can be noted that both the macro- and micro-cracks propagated in a direction perpendicular to the direction of the applied loading. The abrupt drop in the load is explained by the fact that a localized macro-crack has developed in the tie beam specimen. The rate of the drop has decreased compared to when the first crack appeared. In the zone of loads 45 to 47 kN, the reinforcing bar is undergoing plastic behavior. Overall, from the load displacement curve in Figure 15, three macro-cracks can be observed for the normal strength concrete (NSC) tie specimen [60].

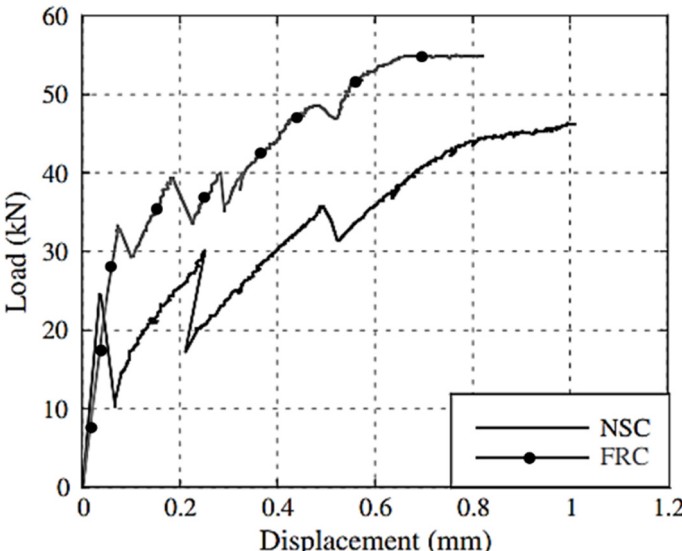

**Figure 15.** Load–displacement curves for tie-beams testing (Adapted with permission from reference [60]).

The tie-beam testing conducted on fiber reinforced concrete (FRC) and normal strength concrete (NSC) tie specimens is studied for mechanical properties under static loading. Figure 15 shows the load–displacement curves for the different tie specimens. The specimens showed similar mechanical behavior compared to each other; however, for the FRC, the sudden drop of force was observed to be less than that of the NSC in Figure 15. The higher forces prior to reaching the yielding of the rebar were due to the high contribution of the concrete reinforced with fiber to the total tensile strength [59,60].

## 5. The Permeability at the SCI

Local properties at the SCI are impacted by external factors and applied loads. Engineering structures over their service life are subjected to various degrees of moisture, wetting and drying, continuous submersion in water, temperature, and different chemicals. Permeability as a term refers to the material's ability to withstand the penetration of gas, water, and chloride under a pressure gradient. It is an essential criterion which affects the durability of reinforced concrete [26,67–69]. Moreover, chemicals such as chlorides and sulfates impact both the reinforcing steel (corrosion) and concrete [23]. Even though the concrete cover helps in protecting the SCI against external environmental factors, it is only efficient for a limited period of time. However, when cracks and gaps in concrete are subjected to harsh environmental conditions there will be no protection of the steel-concrete interface. The increased exposure to aggressive agents will lead to the degradation of the concrete at the interface due to the increase in permeability. The existence of paths in a porous concrete structure allows the passage of fluids, which is responsible for the dynamics of the fluid required time so that the flow reaches the steady state because of low transfer resistance [40]. The voids present at the interface and induced cracks foster the transfer, which significantly impacts the durability in an adverse manner [71–73].

As a summary, regarding the mechanical behavior of the bond strength of the steel and concrete, it was shown that that: There different phases exist that contribute to the mechanical behavior, which are chemical adhesion, mechanical interlocking, and friction resistance. Several factors influence the mechanical behavior at the SCI such as the geometry and diameter of the rebar, the strength grade of concrete, anchorage length, and binder type, but more than one factor may be interrelated with one another. Ryota et al. and results from Zhijian et al. showed that for pull-out testing using ribbed steel in comparison to plain or smooth steel possessed higher bond strength where mechanical interlocking takes place due to the presence of ribs. In push-in testing for ribbed steel at the SCI, only shear failure will occur as the anchorage length was considered to be short compared to the total length of the

specimen. Upon loading, the behavior was at first linear till reaching a maximum load, after that a diminished effect was observed for larger displacement values to reach zero where friction forces are diminished. After studying the mechanical behavior in tie specimens of normal strength concrete and fiber reinforced concrete, several cracking stages were obtained where FRC showed high forces prior to reaching the yielding of the rebar due to the high contribution of the concrete reinforced with fiber to the total tensile strength.

### 5.1. Permeability Test Setup Systems

Permeability systems were developed to evaluate the permeability of a porous material through the flow of fluid or gas at pressure gradient. Three types of testing were conducted on steel reinforced specimens, push-in testing [57], tie beam testing [59,60], and Cembureau constant head permeameter [21]. The push-in testing is implemented in order to examine the gas transfer, affected by the shear stresses at the steel-concrete interface (Figure 16). The properties of the specimen are the same as previously discussed in Section 4.1. Using Darcy's law, the apparent permeability $k_a$ (m$^2$) in the longitudinal direction is calculated using Equation (1).

$$k_a = \frac{Q_i^l}{S} \frac{2\mu L P_{atm}}{(P_i^2 - P_{atm}^2)} \tag{1}$$

where $Q_i^l$ is (m$^3$/s) volumetric flow rate, $S$ is the section perpendicular to the flow, $L$ is the percolation length equal to the interface length of 0.07 m, and $P_{atm}$ (Pa) and $P_i$ (Pa) are the atmospheric and injection pressures, respectively. $\mu$ (Pa.s) is the nitrogen dynamic viscosity. In order to determine the intrinsic permeability $k_v$ (m$^2$) for a laminar flow, Klinkenberg's approach is adopted. This approach is specified in Equation (2), the Klinkenberg equation, as follows:

$$k_a = \left( k_v + \frac{\beta}{P_m} \right) \tag{2}$$

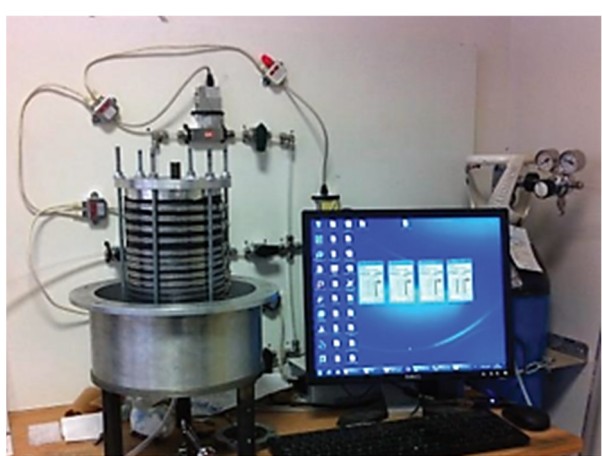

**Figure 16.** Permeability testing setup (Adapted with permission from reference [57]).

The apparent permeability (m$^2$) is assigned with a symbol $k_a$ and average pressure $P_m$ calculated from Darcy's law, $\beta$ (Pa) is a constant called the Klinkenberg coefficient.

Sogbossi et al. [21] conducted an experiment using the Cembureau permeameter to study the transfer properties within plain concrete samples and concrete reinforced with ribbed steel with a diameter of 14 mm. The steel is expected to reduce the voids at the interface due to the presence of ribs that bind firmly with concrete [77]. The Cembureau permeameter is a standardized apparatus that assesses the flow measurement of gas within a porous material. The inlet pressure was regulated from 2 to 5 bar and set pressure level was sustained close to 1% with respect to the pressure selected throughout the measurement of the air flow (Figure 17). The percolation length, which corresponds to the anchorage length, was taken as only 50 mm, so that little time was needed to attain a steady state condition [78].

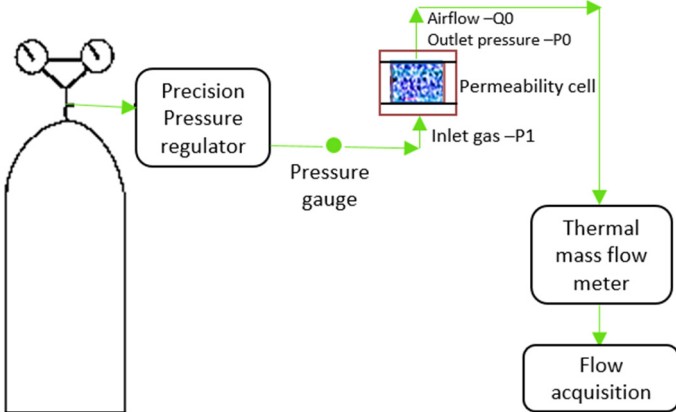

**Figure 17.** Experimental apparatus for permeability testing (Reproduced with permission from reference [21]).

For tie beam specimens (see Figure 8), the permeability test is conducted to study the flow of water through a specimen in the individual tank. Tie specimens are implemented using a steel rebar and concrete as an RC structure, and it is subjected to tensile stresses. In the following permeability system shown in Figure 18, it is completely occupied with water excluding the outlet tank. The water is later set under an initial pressure of 50 KPa or 5 m pressure head from the inlet box, while the output box stays at air pressure. The term water permeability coefficient is defined as the flow of water through the interface and cracks within the concrete matrix. It is assumed that there is a laminar flow that is unidirectional at the steady state. In this case, the equilibrium is indicated where the outflow is equal to the inlet flow within the specimen as specified in Equation (3) Darcy's law.

$$K_w = \frac{QL}{A\,\Delta h} \tag{3}$$

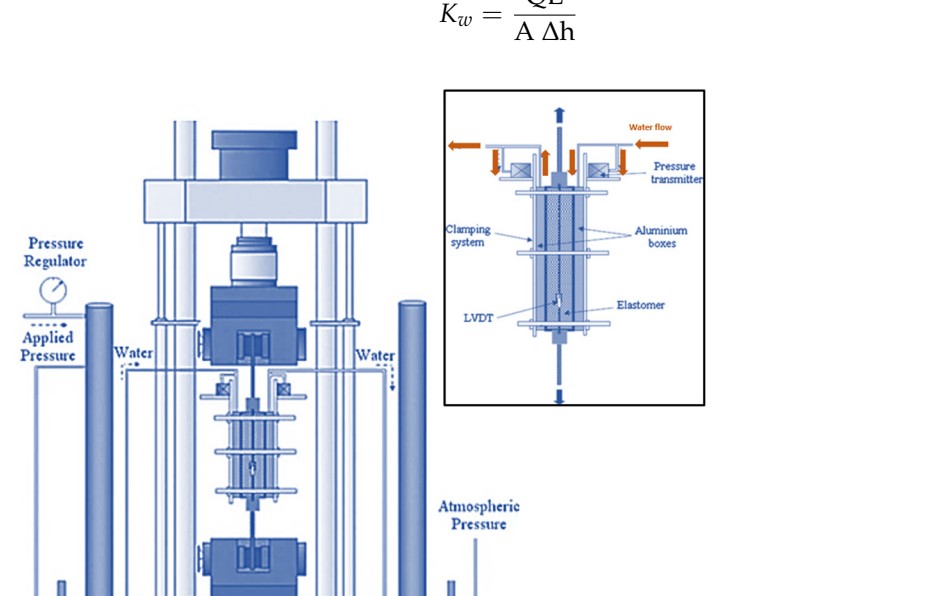

**Figure 18.** Universal testing machine with water permeability device (Adapted with permission from reference [59,60,103]).

The water permeability coefficient (m/s) is assigned with a symbol $K_w$, L is the specimen's thickness (m), the flow rate is Q (m³/s), $\Delta h$ (m) is the drop in hydraulic head along the specimen, and A is the specimen's cross section subjected to flow of water

(m$^2$) [59,60]. In all permeability setup systems, the cells were tightly sealed using rubber so that only the flow would be through the interface.

### 5.2. The Influence of Localized Crack on Concrete Permeability without Reinforcement

Several researchers have conducted splitting tests on plain concrete (unreinforced), which makes it easier to study the relationship between the permeability and crack opening in concrete [79,80]. Before loading, the flow will diffuse within the pores of the concrete, and when the macro-crack is formed, it becomes a preferential path for the fluid and consequently the flow becomes localized in the crack, as shown in Figure 19. There is a direct correlation between crack opening and permeability, hence upon applying load on the disk, the damage will start from the center of the specimen and will propagate vertically to reach the surface of the cylindrical specimen where the permeability increases relatively within this region. Therefore, the application of the splitting test is beneficial to assess the crack opening properties. In general, with the increase in the crack opening, the mechanical properties decrease while there is a significant increase in permeability [79].

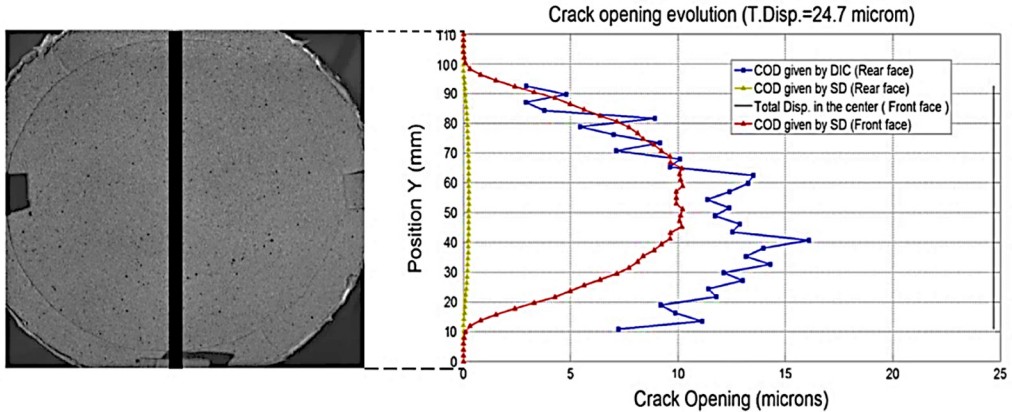

**Figure 19.** Profile of the crack opening and path (Adapted with permission from reference [79]).

Figure 20 shows the relationship between the fracture permeability and mean crack aperture for different diameters and thicknesses of concrete cylindrical specimens. It can be observed that the permeability is not only dependent on the diameter size $d_s$ and thickness $t_s$ of specimens, but also on the mean crack aperture size $\bar{a}$. The fracture permeability increased with the increase in the mean crack aperture size for different geometric configurations [80]. When the concrete sample is in compression at low stress level, the water penetration decreases, and this can evaluate the crack opening displacement (COD) at the middle of the face subjected to compression. Following this stage, there is a slight increase in permeability. The Parallel Plate Model (PPM) is a simple model that describes the water evolution with a crack. The aperture $\bar{a}$ is the distance between the two parallel planes that are smooth. The crack or fracture permeability is described as $kf = \frac{\bar{a}^2}{12}$. However, the theoretical law of cubic flow does not consider the tortuosity, change in aperture, and roughness of the crack, hence a correction factor $\alpha > 1$ was recommended, which is an empirical factor where the corrected water permeability is denoted as: $kf = \frac{\bar{a}^2}{12\alpha}$ [80].

### 5.3. The Influence of Reinforcement on Reinforced Concrete Permeability

The porosity at the SCI is affected by the presence of reinforcement bars in horizontal concrete beams [35]. The permeability increases according to the location of the steel bars at the steel-concrete interface, specifically when this composite material is placed in a direction parallel to the inlet pressure flow. Yuxin et al. [62] found that, for steel bars positioned vertically in reinforced concrete structures, the SCI is uniform around the steel bar since there are no macroscopic defects present. However, the SCI is not uniform for bars embedded horizontally due to different processes such as segregation, bleeding, and settlement of concrete during its fresh state, where voids will form at the interface [81].

This explains the presence of voids that are mainly present at the bottom part of horizontal reinforcing bars for a reinforced concrete structure. This zone is susceptible to the flow of water, gases, and chlorides [11]. To compare the permeability at the steel-concrete interface of reinforced concrete specimens R2, R3, and R5 with reinforcement lengths of 20 mm, 30 mm, and 50 mm, respectively, and plain concrete P with flow rates at the steady-state, the airflow evolution of the apparent permeability with respect to saturation was observed [21]. Figure 21 shows the effect of different saturation levels (Sr) on the permeability of plain concrete and reinforced specimens. The overall decrease in permeability behavior was of comparable trends for any saturation level (Sr). It is observed for saturation (Sr = 0%) that the apparent permeability was $12 \times 10^{-17}$ m$^2$ for plain concrete, and there was a significant rise in the permeability for the case of ribbed reinforced specimen R5 (steel crossing the whole specimen) to $24 \times 10^{-17}$ m$^2$. On the other hand, for Sr = 100% (fully saturated) and regardless of whether the specimen was plain or reinforced, the permeability was approximately zero. In conclusion, the zones that contribute to the airflow are as follows: there is the zone of plain concrete and zone where the defect is formed from the presence of steel rebar at the SCI and from induced micro-cracks causing high permeability.

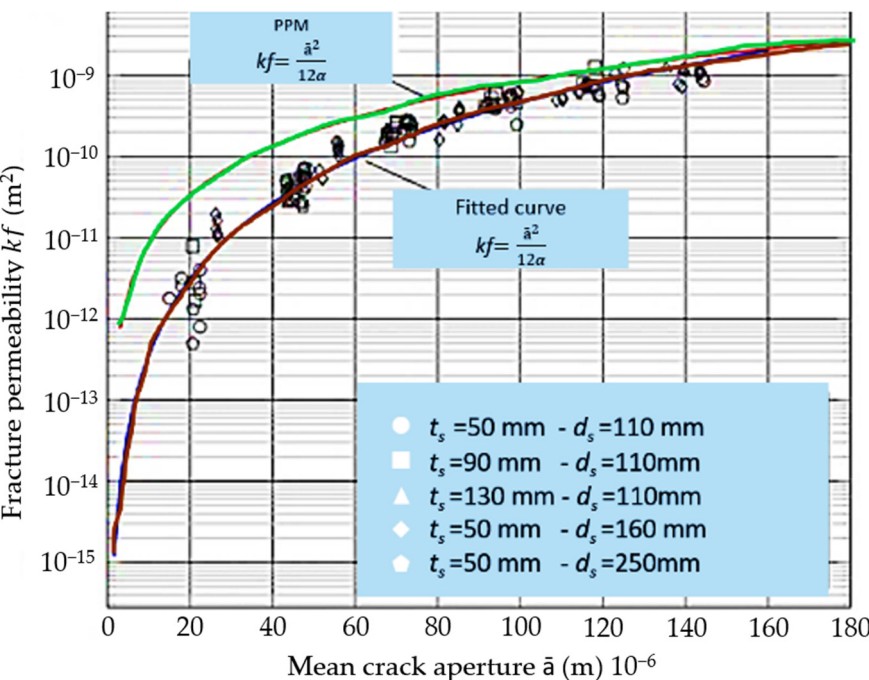

**Figure 20.** Estimation of fracture permeability for concrete cylindrical specimens of different diameters (d$_s$) and thicknesses (t$_s$) green lined curve represents the Parallel Plate Model (PPM) and the red lined curved is the fitted curve (Adapted with permission from reference [80]).

### 5.4. The Influence of Degree of Saturation on Reinforced Concrete Permeability

Studies about the degree of saturation of concrete are mainly focused on the bulk concrete [21], not at the SCI. The degree of saturation at the SCI was studied by Sogbossi et al. [21]. They demonstrated that in reinforced concrete structures when the degree of saturation is high, the only gas flow transfer is at the steel-concrete interface. Figure 21 shows that for saturation (Sr) values between 60 and 80%, small or zero permeability values for plain concrete samples were obtained, whereas the samples were more permeable in the case of ribbed reinforced specimens. This result indicated that micro-cracks were formed at the SCI which exhibit larger openings compared to the pores size of plain concrete. For high saturation degree, the ability of airflow in concrete is reduced. Hence the increase in the concrete saturation level leads to a decrease in the flow of air within the concrete specimen [82–85]. The effect of the steel-concrete interface in reinforced samples is marked by an initial rise in recorded airflow, which later results in greater permeability [21]. Since the SCI is the principal path for the transfer of gas

at high saturation degree, it is required to study the permeability of a structure exposed to different environmental conditions.

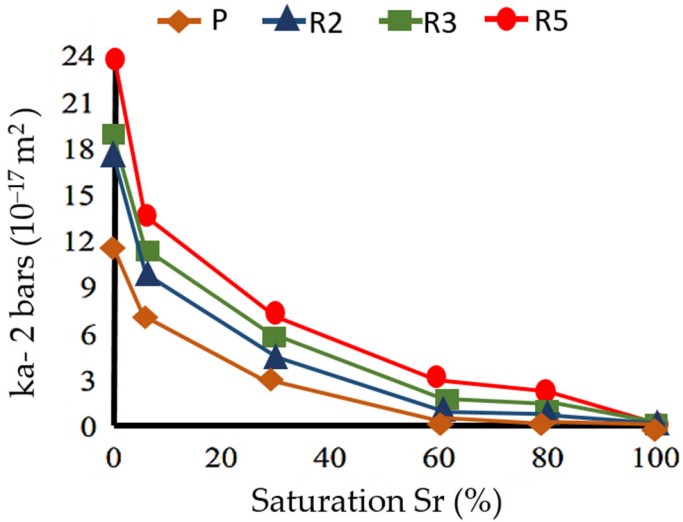

**Figure 21.** Apparent permeability of plain (P) and reinforced specimens R2, R3, and R5 with reinforcement lengths of 20, 30, and 50 mm for different saturation levels Sr% (Reproduced with permission from reference [21]).

*5.5. The Influence of the Concrete Mixture Design on Its Permeability*

The concrete permeability is influenced by the following factors, which are related to the enhancement of the concrete properties in the concrete mixture design:

- Cement paste
- Nominal aggregate size
- Air content (non-air-entrained/air-entrained)
- Water–cement (W/C) ratio
- Degree of saturation
- Concrete matrix strength

The concrete permeability is dependent on the inherent permeability of its components because the hydrated cement paste is more permeable than the aggregates. Tsukamoto [90] indicated that for greater aggregate size (i.e., coarser aggregates), the flow rate of a fluid is reduced for a specified crack opening under mechanical loading compared to finer aggregates, as shown in Figure 22. The permeability of non-air-entrained concrete is reasonably smaller than air-entrained concrete under loading. This may be due to higher void content in air-entrained concrete and under loading, the cracks between voids will allow an easy passage for the fluid [91]. The W/C ratio is a main parameter that affects the characteristics of the concrete microstructure and permeability of concrete within a structure. When studying the gas permeability within a concrete structure having a low and high W/C ratio, results showed that for a small W/C ratio in the concrete, the relatively small pores take up a greater proportion, thus increasing the gas intrinsic permeability [92]. However, the degree of saturation of the structure must also be considered in conjunction with the W/C ratio effect on permeability. The interfacial transition zone (ITZ), the zone between concrete aggregate and cement paste, is widely recognized to have an adverse impact on the permeability of concrete because it is prone to shrinkage and thermal cracking, and it is more porous because the relative size of aggregate is greater than the cement grains where the aggregate acts as a wall affecting the cement grain packing [94]. The matrix strength significantly impacted the water permeability for concrete of normal density. The significant impact of the matrix strength on the water permeability within the concrete can be explained by the fact that after unloading the recovery of the cracks, induced by using feedback-controlled splitting tensile testing, was low for concrete of normal

strength (NSC) compared to high strength concrete (HSC) of high stress–strain linear response [95]. Although regarding the SCI there is research about the microstructure of concrete, macroscopic voids, and steel configuration, however, there is a lack of enough studies about the effect of shear stresses on the permeability at the interface.

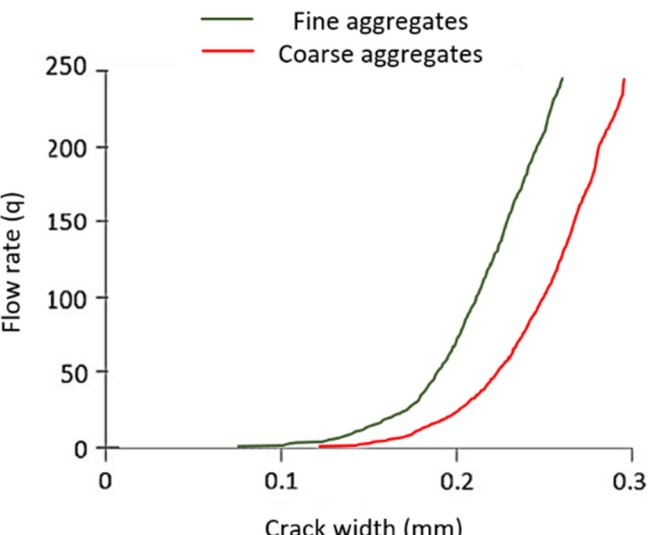

**Figure 22.** Flow rate behavior due to different aggregate size (Adapted with permission from reference [22]).

### 5.6. The Influence of Mechanical Loading on Permeability of Reinforced Concrete

The presence of mechanical stresses comes from applied loading on structures and volume changes affected by environmental factors. This may lead to concrete cracks, separation of constituents, slip in the SCI, and changes on the surface of the steel rebar passive film. Although different environmental exposure conditions relate to local changes, these changes are also time-related, known as the aging mechanism [20]. The above factors affected by loading and environmental conditions will affect the permeability of the structure including the permeability of the SCI [21].

#### 5.6.1. Mechanical and Transfer Behavior Due to Shear Stresses at SCI

An experimental study [57] investigated the evolution of the flow of nitrogen gas at the steel-concrete interface (SCI) when reinforced concrete specimen G1 is subjected to shear loading in a push-in setup. Experimental measurements have been conducted on cylindrical concrete specimens reinforced with a single ribbed steel reinforcing rebar, under different loading conditions. The shear stresses that develop at the interface and gas conductivity represent what might occur in the prestressed reinforced concrete containment buildings of a French nuclear power plant [57]. The permeability prior to loading was determined and considered as a reference permeability. Then the load was applied to induce shear stresses at the SCI in the reinforced concrete cylindrical specimen as shown previously in Figure 7. The specimen was loaded progressively at different loading levels from zero to almost 60 kN and unloaded for residual permeability measurement. In Figure 23 and before any applied loading, the intrinsic gas permeability recorded for specimen G1 was $8.9 \times 10^{-17}$ m$^2$. The difference in the recorded initial permeability of the specimen is mainly governed by the steel defects present at the SCI [21]. At the pre-peak, shear stresses led to cracks formation within the SCI, however, no bond failure has yet occurred, only interlocking took place at this loading stage. Overall, with the increase in loading the interface is adversely affected, so the intrinsic permeability increases [57].

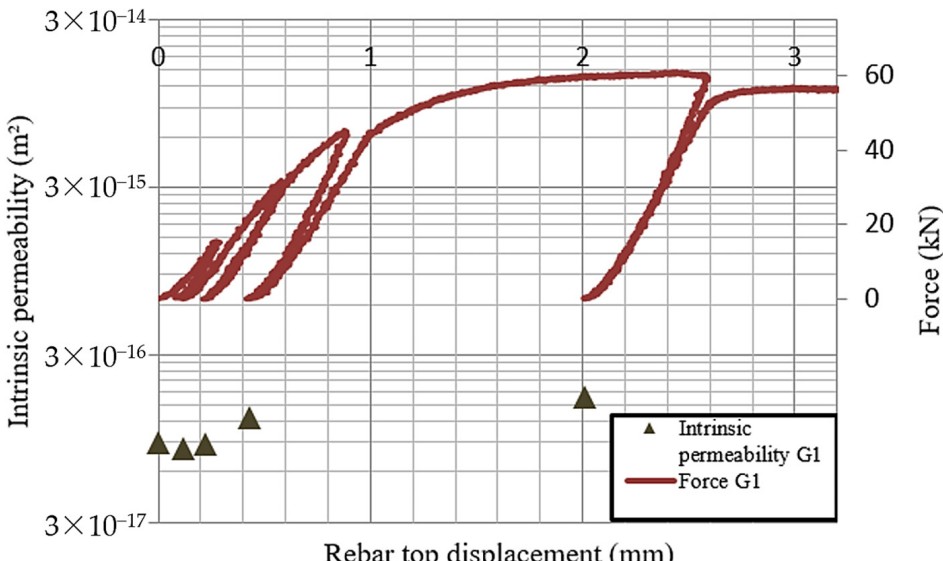

**Figure 23.** Pre-peak phase evolution of the transfer properties for ribbed specimen G1 (Adapted with permission from reference [57]).

For specimen G1, the decrease in permeability to $8.29 \times 10^{-17}$ m$^2$ at a loading of 5 kN was because at certain locations the size of the voids decreased due to compression. Furthermore, in Figure 24 [93], the permeability decreased for loads smaller than 80% peak stress, so the threshold is at 0.8 peak stress. Beyond the threshold, micro-cracks had occurred and caused a permeability increase. However, in Figure 25, it was shown that the applied stress-permeability threshold, to which the permeability began to increase, was found when the applied stress is about 60% of the peak stress [22]. Consequently, it could be highlighted that low load levels might have a positive impact on the overall permeability since the compressive stresses led to less voids [22,105].

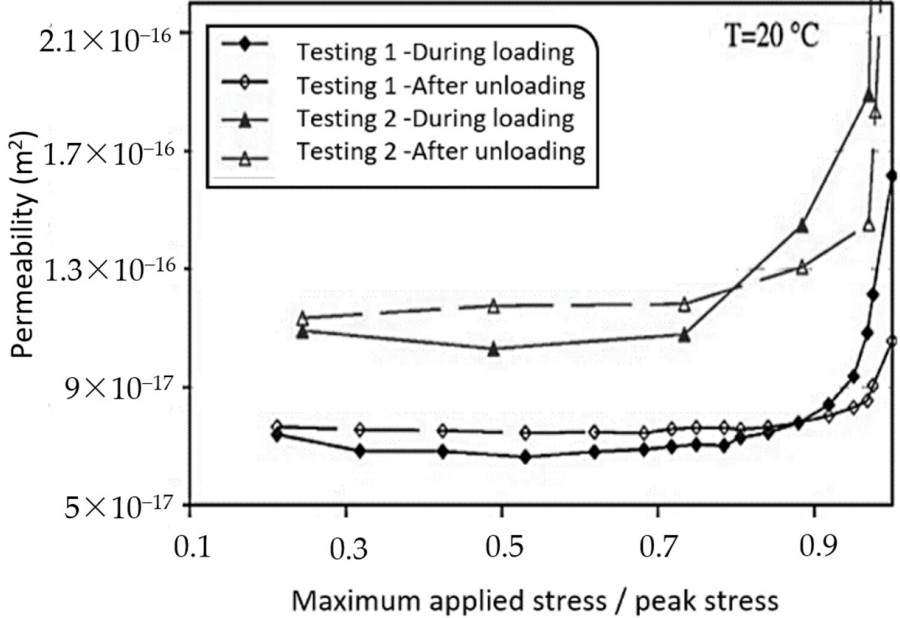

**Figure 24.** Permeability–applied stress relationship (Adapted with permission from reference [93]).

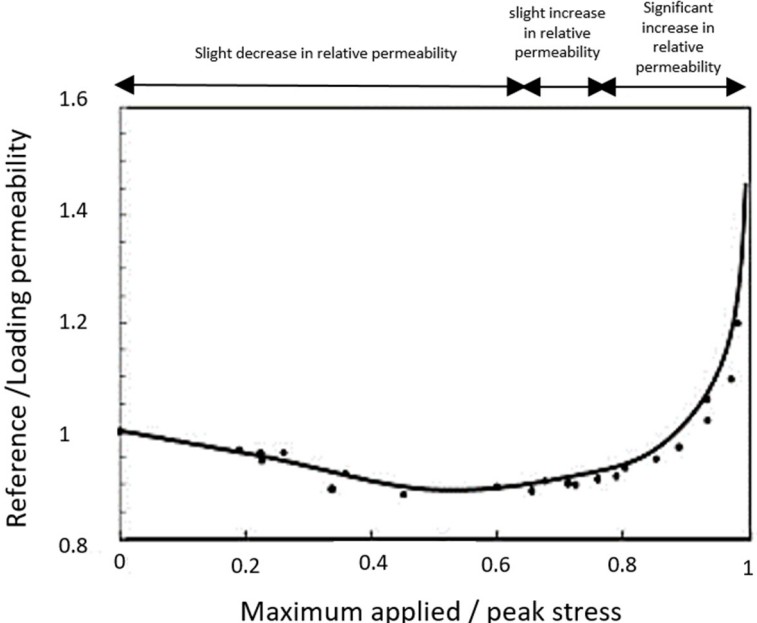

**Figure 25.** $N_2$ gas intrinsic permeability affected by loading in compression (Adapted with permission from reference [22]).

The reduction in transfer properties at the post-peak could be explained by the dilatancy phenomenon of concrete when sheared. Dilatancy occurs when the volume of concrete changes at a certain shear stress that could impact the voids within the structure. The hydraulic behavior between the micro-cracking due to compression and dilatancy phenomena will lead to a high increase in the flow rate for high displacements of the ribbed steel rebar. However, additional research is needed to study the closure of the crack as a consequence of concrete dilatancy while shearing.

For specimen G1 when the applied loading approaches the peak stress, the permeability increases significantly since the concrete links in compression, formed near the steel rebar ribs, were not able to resist the shear stresses, consequently forming inclined cracks with respect to the loading direction at an angle of around 30° [57,58]. Subsequently, the sliding and friction at the interface takes place. As the displacement increases, frictional forces are dominant, and the roughness between the concrete and steel decreases leading to a decrease in the applied load, as shown in Figure 26 [57,59,60]. When the displacement of the top rebar was greater than 40 mm, as shown in Figure 26, the permeability increased rapidly since an interconnected path was formed, which acts as a preferential one for leakage [43]. It should be noted that the rebar displacement at this loading stage is of similar value compared to the initial SCI length of 70 mm in this study. Hence, the remaining anchorage length is relatively short (less than 3 cm), and the flow is percolating mostly in a void zone between the steel and concrete.

5.6.2. The Hydro-Mechanical Behavior of Tie Beam Test

Figures 27 and 28 depict the mechanical behavior of the three reinforced specimens as well as the hydro-mechanical behavior of the five supplementary reinforced concrete specimens, respectively. It demonstrates variation due to heterogeneity of the concrete. The reasons stated are:

- For the individual specimen in each case, the number of macro-cracks developed varies.
- Localized cracking was first initiated by forces between 22 and 28 kN related to the specimen. Upon the development of the first localized crack, a rapid increase in water permeability is detected.

Referring to the following figures (Figures 27 and 28) for the tie specimen, the hydro-mechanical performance is divided into the following phases:

(1) Phase 1: From 0 to 15 kN, the mechanical behavior of the tie specimen is linear elastic. At this stage, the water permeability is nearly constant at $2 \times 10^{-10}$ m/s with slight decrease due to closure of pre-initiated micro-cracks or diminution of pore size.

(2) Phase 2: From 15 to around 27 kN, micro-cracks develop and appear, so there is a slight increase in water permeability by a factor of about 2 ($4 \times 10^{-10}$ m/s).

(3) Phase 3: From around 27 ($\pm 5$ kN) to 45 kN, there is a first crack localization at around 27 kN ($\pm 5$ kN). At this stage, there is a sudden and considerable rise in water permeability reaching around $4 \times 10^{-17}$ m/s. For higher rebar displacement, additional localized cracks (one or two) are also formed. In this case, the localized cracks are considered as privileged pathways for water transfer.

(4) Phase 4: For higher than 45 kN, with the development of macro-cracks, the permeability increases in a regular manner. The rate of increase is low because Poiseuille's regime [79,86–89] has been denoted in the localization of the first crack, and the other openings of the cracks are balanced by the partially closed former cracks [59]. Finally, the water permeability reaches $10^{-5}$ m/s.

The relationship between water permeability and normal stress in reinforcement is shown in the Figure 29. The phases it is divided into are the same as previously mentioned. There is only a difference in the stress of the reinforcement and the required values that would be suitable for structural elements. For the first part, with average normal stress between 0 and 90 MPa, there are micro-cracks smaller than 0.1 mm in size, and the permeability is nearly intact, which is suitable for water-tight structural elements. In the second phase, with stress between 90 and 275 MPa, the maximum opening of the macro-cracks is 0.1 to 0.3 mm, and it is suitable for structural components exposed to harsh environmental conditions. The third phase is for specimens with an average stress between 275 and 400 MPa and a crack opening between 0.3 and 0.4 mm under various environmental conditions. The last phase is where the maximum crack opening is wider than 0.4 mm and is close to the yielding of the reinforcement [59]. This is suitable for the ultimate limit state and not for service limit state.

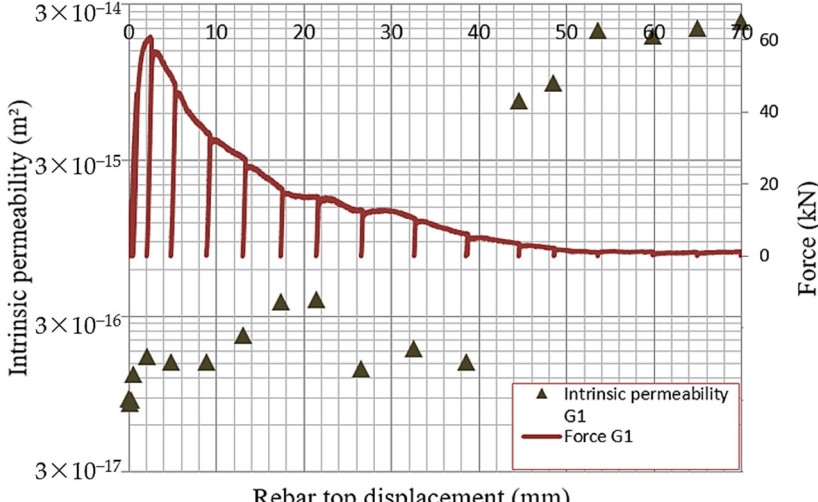

**Figure 26.** Push-in test evolution of the transfer properties for specimen G1 (Adapted with permission from reference [57].

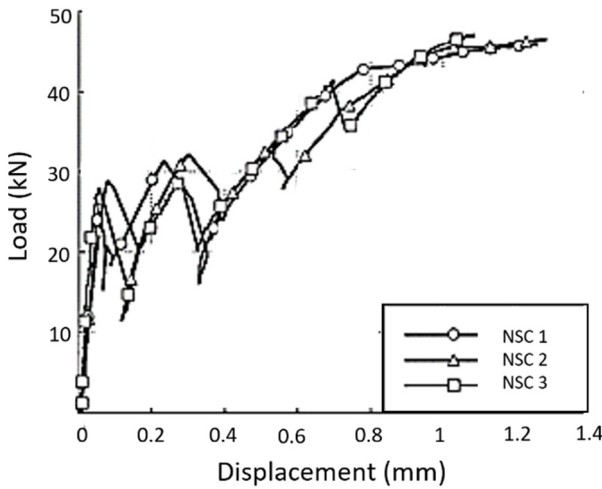

**Figure 27.** Tensile behavior of three repeated tie specimens of normal strength concrete (NSC) (Adapted with permission from reference [59]).

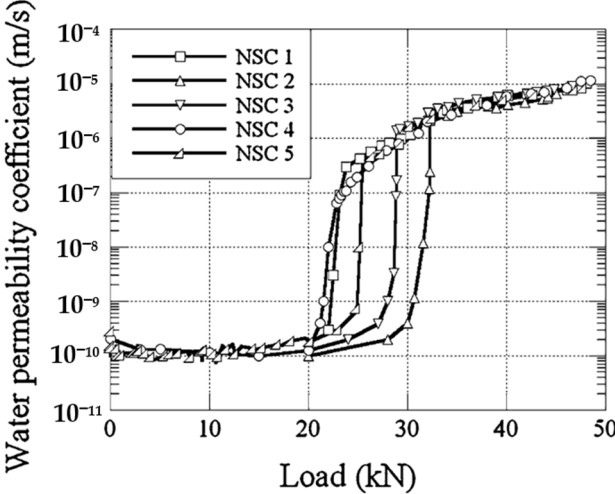

**Figure 28.** Water permeability result with respect to loading for normal strength concrete (NSC) tie specimens (Adapted with permission from reference [59]).

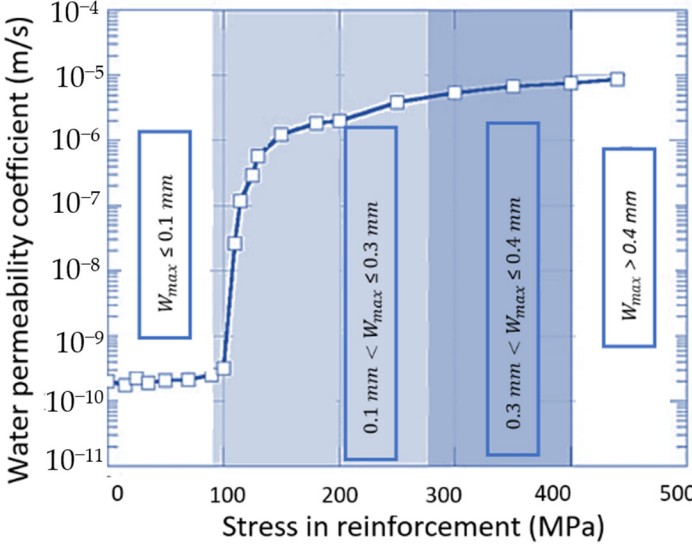

**Figure 29.** Water permeability versus stress in reinforcement (Adapted with permission from reference [59]).

The mechanical behavior of the material influences the permeability at the SCI. The addition of fibers into the concrete mix postponed the initiation of the cracks, so visible branching and thinner cracks will form between the fiber and concrete, leading to an increase in the rigidity and the load cracking recovery of FRC specimens. After studying the permeability of FRC under tensile static loading, there is a difference in the permeability index between the FRC and NSC due to the presence of fibers. Therefore, the permeability of tie FRC specimens was smaller than that of NSC tie specimens. However, in the case of fiber-reinforced concrete specimens (FRC), the water permeability coefficient values have comparable trends with the increase in stress in reinforcement to that of normal strength concrete (NSC). The only difference is that the tie FRC specimens showed lower permeability values by 60% than NSC at the same stress level [60].

Concerning the permeability at the interface, according to the literature review, the following can be noted. Permeability is an important aspect that must be studied to improve the durability and prevent the deterioration of the reinforced concrete structure, which is exposed to aggressive agents such as fluids, chlorides, or leakage of air flow. The permeability was linked to the concrete crack opening from the splitting test. With the increase in crack opening, it will be easier for the flow to travel across the interface, so the permeability will increase. The presence of reinforcing steel at the interface will help in the flow movement of liquid or gas, in which the permeability recorded will be high. The increase in flow rate at the SCI is associated with the constituents of the concrete mix, specifically the aggregate size, whether fine or coarse aggregate, and the crack of the quasi-brittle material (concrete). When studying the flow evolution of gas under push-in loading, shear stresses were developed at the steel-concrete interface leading to an increase in permeability. Overall, with the increase in loading the interface is adversely affected, so the intrinsic permeability increases.

## 6. Use of Models in Understanding the Bond Behavior at the SCI

Before modelling, it is important to have knowledge about the mechanical loading effect on the SCI and induced cracks [70]. The numerical analysis is also dependent on mesh size, shape of the element, and crack size. Hence, there is potential in using a spatially reduced model to better understand the bond behavior between the ribbed steel and concrete at the interface with other factors including the steel bar's anchorage properties and porosity at the interface [4,20]. Therefore, models were implemented to improve the understanding of the bond behavior at the SCI [107,116]. Some of the models used are the finite element model on ABAQUS [1] and mesoscale modelling [63].

A finite element modelling on ABAQUS is conducted by Zhijian et al. [1], this model was established for pull-out testing of reinforced specimens, to simulate the bond at the interface and the characteristics of the steel bar's anchorage. This section covers the crack development at the SCI due to the shear stresses from pull-out testing using finite element modelling. As previously stated, based on experimental results in the first stage, only chemical force was responsible for the slip resistance of the rebar. When the load increases, the chemical adhesion fails, and mechanical interlocking and friction will start taking place [1]. Figure 30a shows the development of concrete crack failure at the SCI. The oblique extrusion force was responsible for the pull-out of concrete near the steel ribs, and it is divided into two components: vertical axial component force and horizontal radial component force. The axial component force comes from the shearing and bending between the steel and concrete. The radial component will be responsible for the presence of tensile stresses around the circumference of the steel-concrete interface. The radial cracks start to form with the increase in load at an angle of 60° with respect to the load direction, applied to the vertical steel bar. In Figure 30b, the in-depth radial failure was double the height of the steel reinforcing rib. The occurrence of this failure mechanism is referred to as ribbed steel shear bond failure [1].

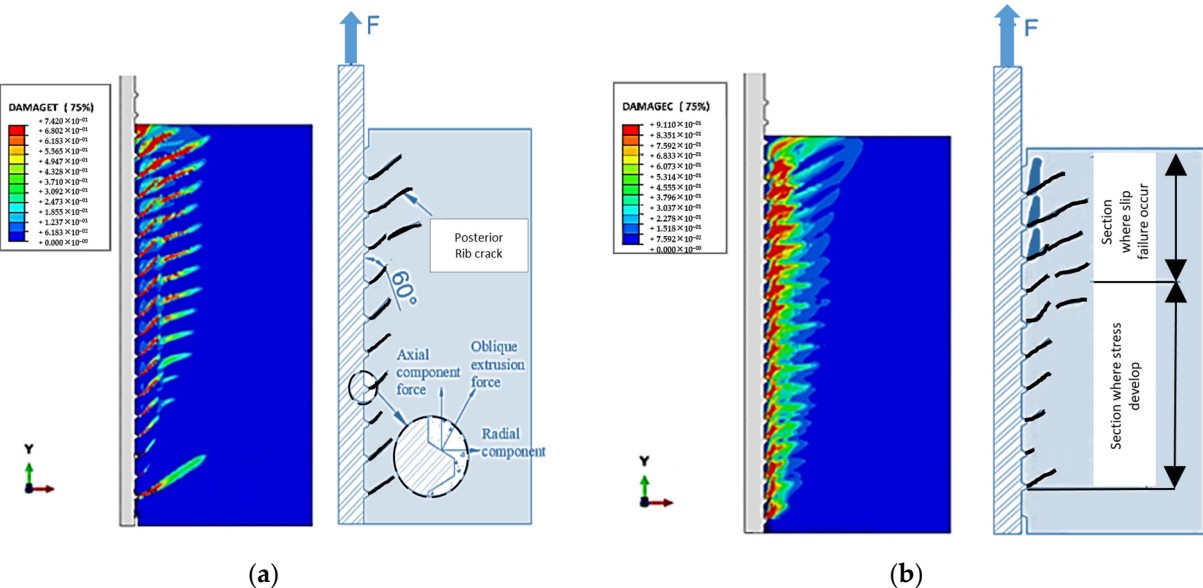

**Figure 30.** (**a**) Development of concrete crack failure at the SCI. (**b**) Concrete failure at the interface [1].

The mesoscale modelling by Mengjia et al. [63] is a realistic model that illustrates how cracks in concrete behave and the bond between concrete and ribbed steel. Unlike typical finite element modelling, in mesoscale modelling, concrete material is modelled as a three-phase composite, composed of coarse aggregates, mortar, and the interfacial transition zone (ITZ). The study is conducted to study the failure mechanism at the SCI with the increase in displacement *s* from 0.4 to 1.5 mm. When loading at initial stage, the damage at the zone of the bond was even around the ribbed steel, then with the increase in loading delamination cracks occurred and continued in an oblique pattern due to the presence of aggregates as shown in Figure 31. In the final stage, ribs fail allowing slippage of the ribbed bar. The damage occurred only at the interfacial transition zone, which is the weaker zone. It was also found that the crack in concrete due to the slippage within the concrete and the ribbed steel is caused by the concrete failure and crushing at the interface facing the ribbed bars.

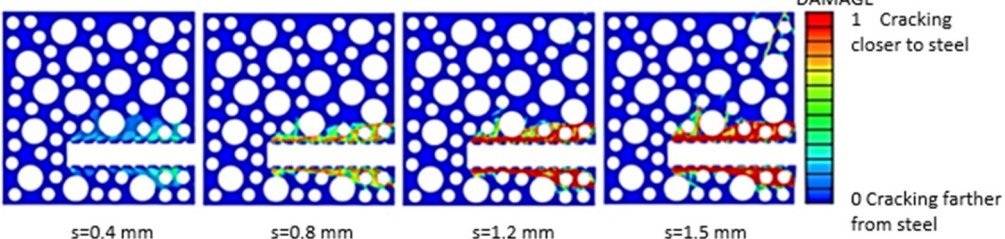

**Figure 31.** Concrete damage process using mesoscale modelling for different displacement(s) [63].

Machine learning (ML), a data-driven technique, offers a cost-effective and efficient solution to properly estimate the bond strength of reinforced fly-ash concrete, taking into consideration all influencing factors. Machine learning models show significant benefits in predicting the ultimate bond strength between fly ash-based concrete and rebars, outperforming empirical models based on pull-out tests. Empirical models can potentially lead to overestimation or underestimation of the specimen result. By relying on machine learning and an empirical model, Shen et al. discovered that for higher slag content, the ratio of anchoring length to rebar diameter (l/d), compressive strength of concrete with fly ash (f'c), reactivity modulus of precursors (RM), and evaluation of the reactive components identified as the ratio of the sum of fly ashes used ($Al_2O_3$, CaO, and MgO to the $SiO_2$ content), all contribute to improved ultimate bond strength [46].

To summarize and indicate whether there is a good agreement between the results from conducting experimental testing and simulated models, the use of models in this section, such as finite element modelling ABAQUS and mesoscale modelling by Mengjia et al., showed good agreement with experimental results. As our world is shifting into artificial intelligence (AI), machine learning models are being used to predict the bond performance behavior at the SCI. However, engineers cannot restrict themselves and rely only on results from models and disregard the importance of conducting experimental testing.

## 7. Knowledge Gaps and Future Research

Structures are subjected to a variety of moisture-related environmental conditions, such as direct sunlight, immersion in water, or being directly exposed to water, and these conditions frequently change over time when the seasons change. This is also relatively applied when subjected to variations in temperature. Moreover, exposure due to chemicals such as sulfates, chlorides, and concentrations of carbon dioxide may differ. Pathological problems or expansion problems will occur in the concrete, namely, alkali silica reaction, carbonation, or sulfate attack from chemical exposure. Hence, the influence of different exposure conditions on the steel-concrete interface needs to be dealt with. The SCI characteristics vary over time as a result of structural loading such as creep effect, deterioration (ageing) of material, and other exposure conditions that change throughout time.

In the literature, other parameters were not taken into account such as temperature, time dependency of the load, and saturation degree where the study was limited to short-term behavior, and durability of the SCI when various exposures exist. In spite of that, the influence of temperature or coupled effect of the mechanical and different types of loading is still not addressed in the literature. As well, extensive research should be focused on the effect of relaxation of steel rebar and prestressed steel strand shape in prestressed concrete structural applications on bond performance at the SCI.

Furthermore, artificial intelligence (AI) must be discussed and included where more research should be applied in the field of an artificial neural network (ANN) using machine-learning-based models developed for the parametric sensitive analysis to evaluate the design of the concrete-to-concrete connections and to evaluate the bond strength at the SCI.

## 8. Conclusions

This study considered the physical properties, effect of loading and flow transfer evolution at the steel-concrete interface (SCI), and the following conclusions can be drawn:

- The microstructure of concrete at the SCI is affected by several factors, including the size of the aggregate, the distance of aggregate particles from the steel bar, the water-cement ratio, the compaction method, the properties of reinforcing steel, and the placement of steel horizontally within the structure. These factors will affect the segregation, flow, hydration, and drying shrinkage of concrete, thus affecting the presence of voids and cracks within this interface.
- The mechanical behavior can be established from the different phases: chemical adhesion, mechanical interlocking, and friction resistance. It is also influenced by the diameter and geometry of the steel rebar (e.g., ribbed or smooth rebar) and the concrete grade. In pull-out testing, the presence of ribs in the reinforcing bar yields higher bond strength compared to plain or smooth steel, which is mainly due to the mechanical interlocking.
- The binder type may have a critical influence in affecting the concrete component of the SCI due to variations in the cement grains' packing, reactivity, and hydration products created. Aluminosilicate minerals such as metakaolin, fly ash, slag, red mud, and steel slag with adequate cement replacement ratios, the microfibers network structure, and excellent hydration ability geopolymer concrete offered various advantages including improved bond behavior over conventional concrete.
- The gas permeability under push-in loading is affected based on three phases. In the first phase, shear stresses are developed at the steel-concrete interface, leading to a

slight increase in permeability. However, in the second phase which occurs at around 0.3 of the peak stress, the permeability is reduced due to compression where the void sizes are decreased. Then, in the third stage, the permeability showed a significant increase with the increase in displacement.

- The presence of fibers in reinforced concrete leads to a higher resistance to cracking. For steel-reinforced specimens, the increase in permeability was influenced by the increase in the reinforcement stress at the interface where cracks developed.
- Machine learning (ML), a data-driven technique, offers a cost-effective and efficient solution to properly estimate and predict the ultimate bond strength at the steel-concrete interface. Ultimately, it is obligatory for the engineer to comprehend the capabilities, efficiency, and limitations of any method of analysis for accurately analyzing and characterizing the physical, mechanical, and transfer behavior at the steel-concrete interface.

**Author Contributions:** Conceptualization, Y.H. and M.E.E.D.; methodology, Y.H.; validation, M.E.E.D.; investigation, Y.H.; resources, Y.H.; writing—original draft preparation, Y.H.; writing—review and editing, M.E.E.D. and J.M.K.; visualization, Y.H.; supervision, M.E.E.D. and J.M.K.; project administration, M.E.E.D. and J.M.K. All authors have read and agreed to the published version of the manuscript.

**Funding:** This research received no external funding.

**Data Availability Statement:** No new data were created.

**Acknowledgments:** The authors thank each individual that has worked on this project for their specific comments and careful reading of the topic.

**Conflicts of Interest:** The authors declare no conflict of interest.

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
