# Peer review of "Physical, Mechanical and Transfer Properties at the Steel-Concrete Interface: A Review"

_buildings, doi:10.3390/buildings13040886_

Round 1

Reviewer 1 Report

This manuscript presents a review of bonding performance at the steel-concrete interface. Thw work is interesting. Related literature is well collected and organized. However, there are still some more work need to be carefully addressed before acceptance

(1)    The title is misleading. It is more like to address steel-concrete composite structures. In reality, it is for steel reinforcement, or reinforced concrete. The title needs modification.

(2)    Although the literature is well collected and organized, it cannot be the only contribution. AI software can do this type job too. There is a need of deep understanding and subsummary of each subtopic. That should be necessary part in a review paper.

(3)    The mechanism of bond-slip was not well explained. For example, the influence of varying parameters on the chemical bonding and mechanical bonding.

(4)    A summary of the outline of this paper is welcome before detailed review on each subtopic. The link of each subtopic should be clearly explained.

(5)    The authors claimed that the main contribution is the influence of shear stress at the surface when subjected to mechanical loadings (Final paragraph in “introduction”). It is not clear. Does it mean shear stress distribution, or shear stress caused deformation, crack development? In the following section 4, it is not clear either.

(6)    There are two 4.2.3 in this manuscript.

Reviewer 2 Report

I believe that the chosen theme is relevant and current. However, I have the following comments/suggestions/questions.

Structure: The research methodology section should be more developed and detailed for an article of this nature. For example, charts and diagrams will improve this section. Also, it is missing a summary of each section, comparing the approaches and results of different authors.

Bibliography: Regarding the "Review" nature of this work, I suggest improving the bibliography with more relevant and recent references in this field of study.

Abstract: Should frame the present work's gaps and main aim.

1.    Introduction: Avoid mass citations like in line 53 [13-19.]

2.    Methodology: This section is too short and slightly explained for a review article. It will benefit from charts and diagrams. It should also justify and frame the following sections.

3.    Physical Characteristics at the Steel-Concrete Interface:

·         Systematic comparison and summary are missing;

·         Subscript numbers, for example: Na2SiO3

·         Why the separation of sections 3.1 and 3.2? Porosity and voids are connected

·         3.4 Binder effect: Is this a physical or a materials’ composition and impact of interaction?

4.    The effect of mechanical stresses:

·         This section is very focused on the results of a specific paper [1], which is not significant.

·         Line 293: Cross-reference Error

·         4.2 should present a summary table analysing the mechanical resistance for different binder type

·         I suggest reassessing the structure of this section to present the data clearly

5.    The Permeability:

·         Table 2 is entirely unnecessary since it only lists factors. Bullets could replace it.

·         This section is very confusing. It will benefit from a reorganisation as section 4.

6.    Use o models:

·         This section focus on very few references, which is not acceptable.

·         Why is the focus only on ABAQUS?

Round 2

Reviewer 1 Report

The authors have improved the manuscript according to reviewer's comments.

Author Response

Thank you.

Reviewer 2 Report

Line 93: Figure Error!

Author Response

Dear reviewer,

Thank you for giving us the opportunity to submit a revised draft of the manuscript “Physical, Mechanical and Transfer Properties at the Steel Reinforcement-Concrete Interface: A Review” for publication in the Journal Buildings MDPI. We appreciate the time and effort that you and the reviewers dedicated to provide feedback on the manuscript.

Reviewer's Comments to the Authors:

Line 93: Figure Error!

Response: Thank you for indicating this error

The error has been corrected upon your request

Sincerely,

Yousra Hachem, Mohamad Ezzedine El Dandachy and Jamal M Khatib